# Radiopharmaceuticals for PET and SPECT Imaging: A Literature Review over the Last Decade

**DOI:** 10.3390/ijms23095023

**Published:** 2022-04-30

**Authors:** George Crișan, Nastasia Sanda Moldovean-Cioroianu, Diana-Gabriela Timaru, Gabriel Andrieș, Călin Căinap, Vasile Chiș

**Affiliations:** 1Faculty of Physics, Babeş-Bolyai University, Str. M. Kogălniceanu 1, 400084 Cluj-Napoca, Romania; george.crsn@gmail.com (G.C.); sanda.moldovean@ubbcluj.ro (N.S.M.-C.); diana.timaru@stud.ubbcluj.ro (D.-G.T.); 2Department of Nuclear Medicine, County Clinical Hospital, Clinicilor 3-5, 400006 Cluj-Napoca, Romania; gabriel.andries@umfcluj.ro; 3The Oncology Institute “Prof. Dr. Ion Chiricuţă”, Republicii 34-36, 400015 Cluj-Napoca, Romania; calincainap@yahoo.co.uk; 4Institute for Research, Development and Innovation in Applied Natural Sciences, Babeș-Bolyai University, Str. Fântânele 30, 400327 Cluj-Napoca, Romania

**Keywords:** PET, SPECT, radiopharmaceuticals, tracers, molecular imaging, review

## Abstract

Positron emission tomography (PET) uses radioactive tracers and enables the functional imaging of several metabolic processes, blood flow measurements, regional chemical composition, and/or chemical absorption. Depending on the targeted processes within the living organism, different tracers are used for various medical conditions, such as cancer, particular brain pathologies, cardiac events, and bone lesions, where the most commonly used tracers are radiolabeled with 18F (e.g., [^18^F]-FDG and NA [^18^F]). Oxygen-15 isotope is mostly involved in blood flow measurements, whereas a wide array of ^11^C-based compounds have also been developed for neuronal disorders according to the affected neuroreceptors, prostate cancer, and lung carcinomas. In contrast, the single-photon emission computed tomography (SPECT) technique uses gamma-emitting radioisotopes and can be used to diagnose strokes, seizures, bone illnesses, and infections by gauging the blood flow and radio distribution within tissues and organs. The radioisotopes typically used in SPECT imaging are iodine-123, technetium-99m, xenon-133, thallium-201, and indium-111. This systematic review article aims to clarify and disseminate the available scientific literature focused on PET/SPECT radiotracers and to provide an overview of the conducted research within the past decade, with an additional focus on the novel radiopharmaceuticals developed for medical imaging.

## 1. Introduction

Over the last decade, the initial focus on medical imaging based on detection and diagnosis has reoriented towards prognosis, tissue characterization, and prediction of treatment efficacy. To this extent, functional imaging, such as positron emission tomography (PET) and single-photon emission computed tomography (SPECT), has become essential in the clinical decision-making process in various fields of medicine. Moreover, hybrid imaging, combining SPECT and PET with computed tomography (CT) or magnetic resonance imaging (MRI), has increased the diagnostic accuracy of both PET and SPECT by the benefit of the morphological information obtained by the CT and MRI scans and the implementation of attenuation correction. PET represents a quantitative imaging tool that appears to surpass the SPECT technique. However, the answer to the highly debated question of which modality will monopolize the nuclear imaging technologies remains unsettled. Traditionally, when compared with SPECT, PET technology provides better image resolution, less attenuation (due to higher photon energy) and scatter artifacts, and, consequently, superior diagnostic capabilities. Two of the most important advantages of PET over the SPECT modality are represented by PET’s higher sensitivity and more robust and flexible tracers, making PET a versatile and powerful tool for clinical and research applications. These advantages, however, come with a high cost burden that limits the availability of PET imaging. Most positron-emitting radioisotopes have short half-lives and require in-house cyclotron production. Therein lies the main advantage of SPECT. Radiopharmaceuticals used for SPECT imaging are cheaper and easy to distribute, and in particular conditions, they present more specific targeting abilities of the biologically active molecules due to the longer half-life of single-photon emitters, allowing for an accurate description of the biological processes at equilibrium in vivo (within several hours, or even days, after the radiolabeled compound’s administration). It is worth noting that the development of the radiopharmaceutical compounds related to distinctive diagnostic and therapeutic targets [1], and therefore used in both imaging modalities, goes hand in hand with the acquisition systems’ development [2].

Figure 1 shows the number of scientific publications over the last 10 years related to radiotracers for PET/CT and SPECT/CT techniques. A clearly increasing trend of publications is observed for both cases, yet with a number of SPECT papers, on average, about seven times smaller.

PET represents the functional imaging technique widely used nowadays for clinical diagnosis of a large variety of diseases, and employs short half-life positron-emitting isotopes, such as carbon-11 (^11^C; t 1/2 = 20.4 min) and fluorine-18 (^18^F; t 1/2 = 109.7 min), for in vivo measurement of biological processes. The technique is also heavily used as a research tool in preclinical studies using animals and for the detection of specific molecules within the human body. In the 1960s, radiopharmaceuticals were already attributed as drugs designed for in vivo diagnosis and treatment applications. A radiopharmaceutical compound consists of: (1) a molecular structure identified as a vehicle molecule and (2) a positron-emitting radionuclide. The radioisotope is attached to the vehicle molecule, also known as ligand, and then injected into the body as a radioactive tracer [3].

Commonly, the vehicle molecules are responsible for the chemical and biochemical reactions within the body; therefore, the connections between vehicle structures and radionuclides are stabilized using chemical linkers. The ligands must present high selectivity and specificity towards their targets. These target sites can be either transporters, enzymes, selected receptors, or antigens. Moreover, the targets can be part of metabolic alterations, tissue hypo-oxygenation, or changes in gene and/or protein expression. However, in pathological conditions, the target’s function might be significantly altered, further affecting the biological interactions between the vehicle part and its target, particularly in tumors, where the receptors, transporters, and enzymes’ expression pathways are heavily affected [3,4].

The PET technique is based on the detection of emitted radioactivity levels of the tracer, normally administrated through an intravenous injection. The radiation doses are comparable to those used in computed tomography (CT) scans [3]. The measurement of glucose consumption rates within different parts of the body is the most common use of PET imaging based on the accumulation of the radiolabeled glucose analogue 18-fluorodeoxyglucose (FDG). Considering that glucose metabolizes at faster rates in malignant tumors when compared with benign ones, this technique is widely used for whole-body scans in order to stage the cancer [4]. Further applications of PET scans include blood flow and oxygen consumption in the brain; tracking of specific neurotransmitters, such as dopamine in Parkinson’s disease; or, in cardiology, evaluation of myocardial viability [3].

A PET radionuclide selection should be considered based on several crucial aspects regarding, first of all, the radionuclide availability, then its physical characteristics, and its radiochemical and radiopharmacological issues [3,4]. With respect to radiochemical considerations, since their primary chemical form is not predisposed to direct labeling reactions, an initial activation step is required for reactive chemical modifications.

A wide array of PET radiopharmaceuticals have been tested and evaluated in clinical trials, targeting a large spectrum of diseases. While all these PET compounds present different compositions in terms of their vehicle molecules (or ligands), they all must follow the same requirements—as imaging agents—with high specificity, high binding affinity, low toxicity, stability (e.g., against different enzymes in plasma), rapid clearance from nontargeted tissue, accessibility at low costs, and permission for clinical usage [4]. The selection or development of a radiopharmaceutical has to meet certain criteria in order to be adequate for an exact biological targeting or disease. Specifically, the radionuclide must have a reasonable half-life, depending on the desired use. In addition, characteristics such as size or charge of the molecule, its specific activity, lipophilicity, stability, and the metabolism of the radiolabeled compounds are directly correlated to the specificity of each biological target. Thus, through quality control tests, aspects concerning the physicochemical, radiochemical, or biological properties are also required [5].

As previously mentioned, alongside the half-life of the radionuclide, the size and mass also play an important role in eliminating the radiopharmaceutical out of the in vivo system. The size of the molecule ensures a better clearance from circulation and has an impact over the in vivo distribution patterns of the radiopharmaceutical. For instance, larger molecules have longer localization time when compared with small molecules, and they cannot be filtered by the kidneys [6]. Additionally, the charges also influence their solubility in different solvents. Noncharged molecules are prone to be more soluble in lipids and organic solvents, whereas radiopharmaceuticals with greater charges present better solubility in aqueous solution.

The radiolabeled compound preparation should be considered in an aqueous solution with a pH as close as possible to the pH of blood. In addition, the ionic strength and osmolality should also be compatible with blood. Their solubility is influenced by their sizes, masses, charges, shapes, and a fundamental physicochemical property, their lipophilicity. Last but not least, lipophilicity has a significant impact on the absorption, distribution, and elimination of drug molecules. For example, neutral lipophilic molecules are usually the only ones able to penetrate the blood–brain barrier (BBB) [7].

Almost all drugs are able, to a certain extent, to bind to blood components. Protein binding depends on the nature of the protein, the concentration of the anions, the charge of the radiopharmaceutical compound, and the pH. Increased lipophilicity encourages nonspecific binding to albumin and other plasma proteins [8]. Metals have a high affinity for proteins, and that leads to a high possibility of ion exchange between a metal complex and a protein. Therefore, the protein binding properties should also be thoroughly studied before clinical use.

In terms of stability, the physicochemical parameters, such as temperature, pH, and light, must be carefully established for the radiopharmaceutical preparation and storage. With regard to the compound’s metabolism, if the radiopharmaceutical compound can be metabolically decomposed, its biodistribution becomes affected because of the mixture of the intact agents and metabolic fragments from the decomposed radiolabeled molecule. The blood metabolism might also alter the delivery of the radiopharmaceutical to the target site. Moreover, the metabolic compounds might get stuck at the target site, and therefore, the relative concentration of the intact radiolabeled molecules, as well as the relative concentration of the metabolic products, must be carefully measured in order to obtain meaningful results [5].

Finally, depending on the concentration of target molecules, a radiopharmaceutical compound must exhibit a proper specific activity (SA). SA is a measure of the number of radioactive probe molecules that are bound to the targeted system. Possible ways of increasing the SA include the purification of the radiopharmaceutical after radiolabeling or the reduction of the quantity of precursor for radiolabeling [5].

On the other hand, SPECT and planar scintigraphy account for almost 80% of all nuclear medicine scans performed worldwide [9]. SPECT radiopharmaceuticals have similar design considerations as described for PET but are based on gamma-emitting radioisotopes, such as ^99m^Tc, ^123^I, ^131^I, ^111^In, ^67^Ga, ^201^Tl, ^81m^Kr, ^133^Xe. While PET has the advantage of higher resolution and sensitivity, SPECT is more accessible and cheaper. PET/CT hybrid imaging accounted for the main limitation of PET, namely, uptake localization. With the adoption of PET/CT imaging in clinical practice, focus has shifted to developing novel PET radiopharmaceuticals. However, as mentioned in recent reviews [9,10], SPECT still plays an important role in nuclear medicine imaging. A wide variety of radiopharmaceuticals are available for SPECT imaging techniques that are integrated into the clinical decision-making process, such as [^99m^Tc]-sestamibi and [^99m^Tc]-tetrofosmin for the diagnosis of cardiac ischemia or [^99m^Tc]-labeled diphosphonates for identifying bone metastasis in breast or prostate cancer.

In recent years, developments in SPECT imaging systems based on new solid-state cadmium telluride and zinc telluride (CZT) crystals and collimator design led to an increase in resolution and sensitivity. Furthermore, advances in radiometal-based radiopharmaceuticals for PET, specifically the successful development of [^68^Ga]-PSMA-11, can be translated to SPECT radiometals, such as technetium [10]. These considerations have rekindled interest in designing novel SPECT radiopharmaceuticals.

As with all medical applications that use ionizing radiation, the benefit of PET and SPECT procedures must be evaluated considering the risks to patient. Dose optimization takes into consideration the administration of the amount of radioactivity that provides images of sufficient quality so as to achieve the relevant clinical information while maintaining the lowest possible radiation dose to the patient. There are different aspects that must be taken into account when deciding the administered dose, such as individual patient physiology and anatomy or the design of the imaging equipment used for the procedure [11]. Average effective doses for nuclear medicine procedures range from 0.3 to 20 mSv with SPECT having generally lower effective doses than PET mainly due to the physical characteristics of the radioisotopes used. For example, the average effective dose for a [^99m^Tc(I)]-sestamibi cardiac rest–stress test (2-day protocol) with an administered activity of 1500 MBq is 12.8 mSv, while for a cardiac [^18^F]-FDG PET scan with an administered activity of 740 MBq, the average effective dose is 14.1 mSv [12]. Hybrid systems increase the radiation exposure by the addition of a CT scan. The additional radiation dose depends on whether the CT scan is used for attenuation correction, localization, or diagnostic acquisitions [13].

The present review paper provides an overview of current and novel PET/SPECT radiopharmaceuticals used in the past 10 years for medical and preclinical applications. The article comprises PET and SPECT radiotracers used in clinical/preclinical oncology for central nervous system imaging, cardiovascular events, bacteria imaging, inflammation and infections, and nonspecific interactions. Imaging using PET/SPECT agents for other diseases were also considered.

## 2. Results

### 2.1. PET Radiopharmaceuticals

#### 2.1.1. PET Radiopharmaceuticals in Oncology

##### [^18^F]-Labeled Compounds

Since ^18^F is more stable as a radioisotope, its labeling has been the most widely used option in the manufacture of PET radiopharmaceuticals. Nevertheless, due to the higher electronegativity of the F atom (4.0) compared with the H atom (2.1), ^18^F labeling exhibits a great impact on the vehicle molecule physicochemical properties. Moreover, the C-F bonds are more stable (in vivo) and stronger than the C–H bonds. Therefore, the inclusion of F in the biological molecule structures implies an extension of their half-lives within the organism, affecting the molecules’ metabolization, biodistribution, and protein-binding kinetics [4].

The gold standard PET radiopharmaceutical, the [^18^F]-fluorodeoxyglucose ([^18^F]-FDG) compound, is being taken up by the cancerous cells relying on the enhanced metabolic and glycolytic rates within the intracellular matrix [14]. However, in 2015, a meta-analysis study conducted by Deng et al. concluded that ^18^F-FDG uptake in cancer patients shows just a moderate correlation to cancerous cell proliferation [15], as its uptake has also been observed in other infectious and/or inflammatory diseases. The “molecule of the 20th century” was conceptualized in the early 1970s, synthesized in 1978, and initially applied in neuroimaging [4].

As a short overview, with [^18^F]-radionuclide’s half-life of 109.7 min, ^18^F-labeled PET compounds (Appendix A) are used in prostate cancer, breast and gynecologic cancers, lung cancers, glioblastoma, hepatocellular carcinoma (HCC), solid malignancies, head and neck cancers, colorectal and pancreatic cancers, and abnormal mass tissues known as neoplasms [4].

Radiolabeled amino acid (AA) PET radiopharmaceuticals are based on endogenous molecules (originate within the body), and are widely used in oncology imaging [16], particularly in glioma imaging due to a lower background uptake (in the brain) when compared with ^18^F-FDG [17].

Examples of ^18^F-labeled AAs include L-6-[^18^F]-fluoro-3,4-dihydroxyphenylalanine ([^18^F]-FDOPA), 2-[^18^F]-fluoroethyl-tyrosine ([^18^F]-FET), 4-fluoroglutamine ([^18^F]-FGln), (4*S*)-4-(3-[^18^F]-fluoropropyl)-L-glutamic acid ([^18^F]-FSPG), trans-1-amino-3-18F-fluorocyclobutanecarboxylic acid ([^18^F]-FACPC), and anti-1-amino-3-[^18^F]-fluorocyclobutane-1-carboxylic acid ([^18^F]-FACBC). Liu et al. reported several ^18^F-labeled AAs developed by replacing the carboxylate group (–COO−) with a radiolabeled isosteric trifluoroborate (BF3−) group [18]. Moreover, Britton and coworkers endorsed a method to produce ^18^F-labeled AAs that can track glioblastoma and prostate adenocarcinoma xenografts (e.g., [^18^F]-fluorothymidine ([^18^F]-FLT)), through electrophilic radiofluorination of inactivated C–H bonds in hydrophobic amino acids [19,20]. The [^18^F]-FLT compound enters cancer cells through specific nucleoside transporters and, once inside the cell, is being phosphorylated by thymidine kinase-1 and ultimately confined within the cell. Furthermore, it is widely used to predict chemotherapy/radiotherapy responses for patients diagnosed with lung, breast, and prostate cancers [4].

Among other standard [^18^F]-labeled radiotracers presented in Appendix A, dozens of [^18^F]-estrogen analogue compounds have been prescribed. The [^18^F]-fluoroestradiol ([^18^F]-F-FES) and [^18^F]-fluorodihydrotestosterone ([^18^F]-FDHT) are an example of steroid derivatives able to target androgen and estrogen receptors, respectively [21].

Solid malignancies are accumulated abnormal masses of tissue that may be benign (not cancerous) or malignant (cancerous tissue). The malignant phenotypes can be correlated with sarcomas, carcinomas, and/or lymphomas. In this context, several PET radiotracers used in clinical imaging of solid malignancies were reported in the last decade, such as [^18^F]-EF5, [^18^F]-FMISO, [^18^F]-FAZA, [^18^F]-HX4, [^18^F]-FETNIM, and [^18^F]-FDG [22,23,24,25,26,27,28,29].

These radiopharmaceuticals are small molecules used in lung cancer and for the investigation of the hypoxic sites. The modification in hypoxia permits a better local control and post-treatment prognosis [30].

A salt vector type of tracers (e.g., [^18^F]-NaF [31] and [^18^F]-F-choline [32] is used for the detection of metastatic bone stages of diseases (osseous lesions), and for early detection of the biochemical recurrence in prostate cancer, where ^18^F-choline shows elevated diagnostic accuracy, among others (e.g., [^18^F]-fluciclovine and [^18^F]-PSMA) [32].

The use of dynamic galactose-analogue 2-[^18^F]-fluoro-2-deoxy-d-galactose ([^18^F]-FDGal) PET, fused with CT scans, requires arterial blood sampling from the radial artery and enables the noninvasive in vivo measurement of metabolic function [33]. However, Horsager et al. demonstrated that metabolic liver function (with or without liver disease) can be measured using [^18^F]-FDGal PET/CT but without arterial blood sampling. The method involved extracting an image-derived noninvasive input function from a volume of interest [34]. For the detection of extrahepatic HCC metastases, the use of ^18^F-FDGal PET/CT was found to be significantly superior to both standard clinical PET scans with contrast-enhanced CT and [^18^F]-FDG. Moreover, in the same study conducted by Bak-Fredslund et al., it was observed that [^18^F]-FDGal PET/CT is able to detect (previously) unknown extrahepatic metastases in HCC patients [35].

It is well known that [^18^F]-FDG is the most commonly used PET tracer in oncologic diseases; however, in more complex anatomical regions, [^18^F]-FDG presents a few limitations due to its physiological uptake in normal conditions, leading to challenging image interpretation. The small sizes of the anatomical components at the head and neck region and the inflammatory processes that occur in head and neck squamous cell carcinoma (HNSCCs) patients might be a cause of misleading PET results [36], such as increased [^18^F]-FDG uptake due to inflammatory cells’ activation [37]. In contrast, Helsen et al. evaluated the diagnostic performance of FDG-PET/CT after radiotherapy treatment in patients with HNSCCs, and concluded that [^18^F]-FDG can detect residual disease 11–12 weeks after therapy. However, they also suggest an optimal re-evaluation of 10–12 months post-treatment in order to detect possible late recurrences [38].

In 2020, a clinical trial report conducted by Schöder et al. aimed to determine the feasibility of the [^18^F]-PARPi tracer in PET scans for patients diagnosed with head and neck cancer. [^18^F]-PARPi was seen to be well tolerated by patients without imposing any safety concerns, and was able to detect not only the primary lesions but also the metastatic spots with a higher retention in tumor cells, when compared with healthy ones [39]. Taking into consideration that PARP proteins are overexpressed in malignant structures at the oral and oropharyngeal level, the molecular imaging of PARP is possible with only little unspecific uptake in normal tissue [40,41], and therefore, the [^18^F]-PARPi tracer serves as a clinical tool for oropharyngeal cancer patients [42]. Moreover, it is important to note that this tracer is able to cross the blood–brain barrier, presenting high uptake levels in brain cancers [43].

Since the biggest challenge in targeting cancer is directly correlated with the ligands’ ability to specifically recognize the cancer cells, bombesin (BN) receptors have shown to increase on-site delivery mechanisms and present a promising approach for tumor targeting [44,45]. With this in mind, due to their overexpression in different type of cancers, [^18^F]-BAY 864367 is a peptide-receptor-based radiolabeled ligand with elevated affinities for BN receptors in patients with prostate cancer [46]. However, further clinical trials are required for bombesin receptors to be considered as a potential tool for diagnosis and/or therapy [47].

##### [^11^C]-Labeled Compounds

The ^11^C radionuclide emits with a maximum energy of 960 keV and has a half-life of only 20.4 min. The substitution of the carbon with a positron-emitting isotope in biological structures makes possible the development of specific labeled compounds, enforcing identical biochemical and pharmacological/pharmacokinetic properties to those of the natural molecules [48,49]. The short radioactive half-life of ^11^C involves that the radiopharmaceuticals labeled with this radionuclide do not require substantial radiation exposure, and allows the conduct of multiple studies for a short time interval and in the same individual. In addition, despite the fact that carbon-11 has a short half-life, it is also long enough for synthesis and purification. However, due to its radioactive decay, the radiosynthesis time should be kept as short as possible [50]. In terms of their dynamical properties, the biophysical characteristics of ^11^C-labeled compounds, such as raclopride, FLB, and SCH, are also described within the literature [51].

The manufacture of ^11^C-labeled compounds (Appendix A) requires the availability of a cyclotron facility near the hospital where the study is to be performed, since it must be developed on-site at the time of use [48,52]. Carbon-11 decays to stable boron-11 mostly by positron emission (99.79%) and, to a lower extent, by electron capture (0.21%) [48]. According to Dahl et al., a radiotracer with a molar activity greater than 40 GBq/μmol is acceptable for most PET experiments [52]. Carbon-11 can be produced with a high molar activity in the range of 40–750 GBq/μmol at the end of synthesis [53]. [^11^C]-carbon dioxide, [^11^C]-methane, and [^11^C]-fluoroform are examples of secondary [^11^C]-synthons that have been recently developed from the primary cyclotron-produced [^11^C]-precursors [54,55,56,57,58].

Regarding the [^11^C]-radiopharmaceuticals’ classification based on their vectors, the standard [^11^C]-choline tracer is known to be taken up by cancer cells during proliferation in patients diagnosed with prostate cancer [4,59]. Since acetate is an essential substrate in cell energy and is quickly metabolized into acetyl-CoA in human cells, another salt vector-based compound is [^11^C]-acetate [60] widely used for general cancers [4,60]. The tracer was employed in urological malignancies, renal cell carcinoma, and bladder cancer. Moreover, several studies reported that [^11^C]-acetate PET has also been considered and used in other types of malignancies [61], such as lung carcinomas and brain tumors, and that this tracer is able to detect rare tumors (e.g., multicentric angiomyolipoma of the kidney, thymoma, and cerebellopontine angle schwannoma) [60].

In the context of prostate cancer, [^11^C]-acetate cannot accurately distinguish between benign prostatic hyperplasia and prostate cancer, presenting comparable uptake in both conditions [62]. Another study conducted by Jambor et al. showed high sensitivity (of 88%) but low specificity (41%) of the tracer uptake in 36 patients with untreated and nonmetastatic prostate cancer [63]. In contrast, other studies [64] reported higher uptake affinities of [^11^C]-acetate in tumor cells than in normal prostate tissue. However, potential false-positive uptakes might also account for the inflammatory effects within the cancer cells. In 2012, Schöder et al. supported these findings due to the large number of false-positive lymph nodes observed in their study, generated by chronic granulomatous disease (CGD) [65]. For the assessment of pelvic lymph nodes’ involvement [66,67,68,69], several studies reported either acceptable sensitivity (68%) and specificity (78%) of [^11^C]-acetate uptake [66] or lower patient-based sensitivity of only 38% for lymph node detection [67]. Intriguingly, other two studies reported [^11^C]-acetate as a suitable predictor of lymph nodes’ involvement [68] and a pelvic lymph node’s detection with higher sensitivity (90%) and specificity (67%) [69]. As a predictive biomarker, [^11^C]-acetate uptake was associated with higher prostate-specific antigen velocities [70]. Last but not least, a study conducted by Spick et al. showed comparable conventional bone scans and [^11^C]-acetate PET on patient-based analysis, suggesting that PET imaging using this tracer can accurately assess distant (bone) metastatic involvements [71].

Another small-molecule-based radiotracer, known as [^11^C]-erlotinib, is heavily used nowadays in PET scans for lung carcinomas and colorectal cancer [4]. In 2016, Bahce et al. studied the effects of erlotinib (the medication used to treat non-small-cell lung and pancreatic cancers) treatment on [^11^C]-erlotinib uptake in lung cancer patients [72]. Five years later, Petrulli and coworkers showed that, among subjects with non-small-cell lung cancer (NSCLC) and various epidermal growth factor receptor mutations, the kinetic properties of the tracer varied substantially. In addition, they also implemented a novel scanning protocol that highlighted the pronounced heterogeneity of (non-small) CLC and its impact on [^11^C]-erlotinib [73]. In this context, a study conducted by Yaqub et al. aimed to find the most advantageous pharmacokinetic model for [^11^C]-erlotinib uptake’s quantification in patients diagnosed with NSCLC [74].

##### [^124^I]-Labeled Compounds

^124^I-labeled compounds (Appendix A) are used for both imaging and therapy, as well as for ^131^I dosimetry, due to the long half-life (of 4.18 days) and physical properties of the positron-emitting isotope of iodine [75]. I-124 is also an engaging radionuclide for mAbs development as potential immuno-PET imaging pharmaceuticals [76] and well-established methodologies for radioiodination [77,78]. In patients diagnosed with thyroid and parathyroid cancer, ^124^I can be labeled against mAbs, peptides, small molecules or proteins for tumor imaging [79,80,81], or single molecules such as metaiodobenzylguanidine (MIBG), amino acids, and fatty acids for heart and brain disorders’ investigations, including studies on neurotransmitter receptors and photodynamic therapy [82].

Moreover, iodine radioisotopes have also been widely used as theranostic agents in thyroid cancer [83]. Although there are a total of 37 known iodine isotopes that undergo radioactive decay, the conventional iodine radionuclides used for preclinical and clinical applications are ^123^I, ^124^I, ^125^I, and ^131^I.

Several ^124^I-label-based small molecules have been developed by nucleophilic and electrophilic substitution reactions and, afterwards, tested for various targets. For instance, a few pharmaceuticals are based on small molecule, such as ^124^I-MIBG for adrenergic activity; ^124^I-IAZA and ^124^I-IAZG as hypoxia agents; ^124^I-dRFIB, ^124^I-IUdR, and ^124^I-CDK4/6 inhibitors of cell proliferation; ^124^I-hypericin targeting protein kinase C; and ^124^I-FIAU against herpes virus thymidine kinase [84]. ^124^I-IPPM compounds target opioid receptors, and ^124^I-IPQA engages EGFR kinase activity and ^124^I-labeled-6-anilino-quinazoline derivatives’ irreversible bind to EGFR. In addition, ^124^I-purpurinimide derivatives are also used as tumor imaging agents [75,84].

A large array of ^124^I-labeled compounds based on antibodies, nanobodies, antibody fragments, and proteins have been intensively used for molecular imaging of thyroid cancer, breast cancer, colorectal cancer, renal cell carcinoma, ovarian cancer, neuroblastoma, and so forth. In addition, immuno-PET imaging tracers have been considered for the detection of tumors (over)expressing human EGFR. As an example, ^124^I-labeled ICR12—a rat mAb recognizing the external domain of the human c-Erb B2 protooncogene—was evaluated in patients diagnosed with breast cancer, while another study was focused on two ^124^I-labeled mAbs (MX35 and MH99) evaluated in rats with subcutaneous human (SK-OV-7 and SK-OV-3) ovarian cancer xenografts [75], where, as a result, small (about 7 mm) ovarian cancer nodules were identified using PET imaging.

For the development of anti-HER2 targeting pharmaceuticals, a ^124^I-labeled trastuzumab compound and a small (7 kDa) scaffold protein were evaluated and compared. As a result, the total uptake of trastuzumab in tumors was higher when compared with the uptake of ^124^I-labeled scaffold protein, despite the fact that tumor-to-organ ratios were significantly higher for ^124^I-labeled 7 kDa scaffold protein due to its higher and faster clearance from blood and normal tissues [75].

Moreover, micro-PET imaging and biodistribution studies of ^124^I-trastuzumab were considered in order to examine the compound’s specificity in HER2-positive and HER2-negative mouse models, and as a result, the authors observed elevated tumor uptake levels of ^124^I-trastuzumab when compared with ^124^I-IgG1 in HER2-positive mouse models. In addition, for patients diagnosed with gastric cancer (in metastatic stages), PET/CT images confirmed higher uptake specificity levels of ^124^I-trastuzumab than the uptake levels of [^18^F]-FDG [85].

In 2021, Kumar et al. reported in their review paper that for targeting the CD20 antigen in mice models and CD20 expressing murine lymphoma, cys-diabody (cDb) and cys-mini body (cMb) based on rituximab and obinutuzumab (GA101) were radiolabeled with ^124^I. The use of GA101-based imaging pharmaceuticals, such as ^124^I-GAcDb and ^124^I-GacMb, resulted in higher-contrast immuno-PET images of B-cell lymphoma, when compared with the rituximab-based tracers [75,86].

Other studies have reported novel heavy-chain antibodies (HCAbs) labeled with ^124^I [87] and the development of humanized IgG mAb labeled with ^124^I isotopes [88] for targeting cell death ligand-1 (hPD-L1), known to activate specific (T) cells associated with multiple malignancies. Moreover, fully human ^124^I-labeled IgG monoclonal antibodies have also been tested and used as theranostic agents [89,90,91]. For targeting, biodistribution and safety assessments of ^124^I-labeled compounds, Zanzonico et al. evaluated and characterized (using quantitative PET) the ^124^I-huA33 specific tumor targeting in patients diagnosed with colorectal cancer [92].

In clear cell renal cell carcinoma (ccRCC); carcinomas of the uterine cervix, kidney, esophagus, lung, breast, colon, and brain; and hypoxic solid tumors, the transmembrane carbonic anhydrase IX protein is being overexpressed due to hypoxic conditions of the cancerous microenvironment; therefore, the protein is considered a reliable biomarker of hypoxia [75]. Within this framework, ^89^Zr-labeled cG250 (girentuximab) compounds (as an alternative to ^124^I) were evaluated in ccRCC xenograft models in mice. These studies concluded that, compared with ^124^I–labeled cG250, ^89^Zr-labeled cG250 showed higher uptake, greater retention, and superior PET image quality due to its elevated trapping rates inside the tumor cells [93,94]. However, the cG250 compounds have been labeled with a variety of radionuclides (such as ^124^I, ^111^In, ^89^Zr, ^131^I, ^90^Y, and ^177^Lu), and are lately most extensively investigated as CA-IX (targeting *Carbonic anhydrase*) theranostic pharmaceuticals [95,96,97,98].

In addition, for prostate cancer patients, the well-known and highly debated prostate-specific membrane antigen (PSMA) is already overexpressed in the early stages of the disease [99]. A few studies have reported that ^124^I-capromab compounds suggested relatively lower tumor uptake, with an even lower uptake of ^111^In-capromab in LNCaP xenografts [100]. However, in 2019, Frigerio et al. demonstrated that the uptake of the ^124^I-labeled anti-PSMA single-chain variable fragment (scFv) is very high and specific for PSMA-positive cells [101]. Between 2013 and 2016, a humanized mAb (J591) that binds to the extracellular domain of PSMA was considered and investigated for both imaging and therapy purposes [102,103,104,105,106]. As a result, it has been shown that ^124^I- and ^89^Zr-labeled J591 compounds exhibit comparable internalization rates in preclinical prostate models [106], further implying that prostate cancer theranostics using ^177^Lu- and ^124^I- or ^89^Zr-labeled J591 are not only feasible, but might also involve superior targeting rates of bone lesions relative to conventional imaging modalities [75].

In another study conducted by Knowles et al., the ^124^I- and ^89^Zr-labeled prostate stem cell antigen (anti-PSCA) A11 minibodies were evaluated and compared for quantitative immuno-PET imaging of prostate cancer. Here, the nonresidualizing ^124^I-labeled minibody presented lower tumor uptake than the residualizing ^89^Zr-labeled minibody, although the ^124^I-labeled tracer reached higher imaging contrast due to lower nonspecific uptakes and greater tumor-to-soft-tissue ratios [107]. The same authors compared the ^124^I-labeled A11 minibody imaging with [^18^F]-fluoride bone scans in order to analyze the disease’s progression and response to therapy, and observed that ^124^I-labeled A11 minibody presented higher sensitivity and specificity, contrasting the [^18^F]-fluoride bone scans in detecting the xenografts at all time points [108].

In 2018, Tsai et al. and Zettlitz et al. confirmed, using dual-modality immuno-PET/fluorescence imaging, that the labeled ^124^I-A11 cMb-Cy5.5 presented specific targeting to both 22Rv1-PSCA and PC3-PSCA xenografts in mice, and that the fluorescence imaging signal was strong from both 22Rv1-PSCA and PC3-PSCA tumors, when compared with non-PSCA-expressing tumors [109]. Similarly, the dual probe A2 cys-diabody (A2cDb)-IR800, also targeting PSCA, labeled with ^124^I (124I-A2cDb-IR800) ensued high-contrast immuno-PET images [110].

Last but not least, a few studies have demonstrated that the ^124^I-codrituzumab compound, an antibody targeting glypican 3 (a cell-surface glycoprotein overexpressed in cancerous tissue), is able to detect tumor localization in most patients with HCC [75,111].

##### [^89^Zr]-Labeled Compounds

Due to the fact that ^89^Zr is thermodynamically stable and kinetically inert, can be easily produced within a few hours, and has a long half-life of 3.3 days [112], the use of ^89^Zr tracers represents a promising approach for evaluating the in vivo distribution of monoclonal antibodies in cancer therapies [113,114]. Therefore, a large number of studies have focused on the feasibility of ^89^Zr immuno-PET imaging, with further investigations on the effectiveness of radio-immunotherapy and imaging of target expression, detection of targeted tumors, and monitoring of anticancer chemotherapies. Additionally, ^89^Zr-labeled radiopharmaceuticals have applications in nanoparticle imaging and cell tracking [113]. 

According to Yoon et al. [113], among all synthesized ^89^Zr-labeled antibodies, trastuzumab [115,116,117] is the most frequently employed antibody, followed by bevacizumab [118,119,120,121,122,123,124,125], cetuximab [126,127,128], and rituximab [129,130]. Similarly, the most frequently engaged targets are human epidermal growth factor receptor 2 and 3 (HER2, HER3), epidermal growth factor receptor (EGFR), vascular endothelial growth factor A (VEGF-A), cluster of differentiation (CD) 8 and 20, and PSMA [113].

The antibody-based [^89^Zr]-tracers (Appendix A) are extensively used in prostate cancer [103,131], pancreatic cancer [132,133], bladder cancer [132,134], gastrointestinal adenocarcinoma [127,135,136,137,138,139] and renal cell carcinoma [95,98,137,140,141,142,143], solid malignancies [119,121,144,145,146] and cell lung carcinomas [121,144], and breast cancer [116,129,134,147,148].

Several studies have investigated the feasibility and usefulness of an ^89^Zr radiolabeled compound with J591 monoclonal antibody (mAb)—known to bind the PSMA extracellular domain—in identifying tumor foci [103,104,149,150,151]. In 2019, a few studies assessed the pharmacokinetic and biodistribution properties of an ^89^Zr-labeled DFO-MSTP2109A antibody [152] and confirmed the tracer’s excellent uptake in both soft tissue and bone lesions.

For patients diagnosed with breast cancer, studies reported several treatment and/or diagnosis options based on an ^89^Zr-labeled compound, highly dependent on the (human) epidermal growth factor receptor 2 (HER) assessment [115,116,118,144,153,154,155,156,157,158]. Moreover, in 2013, for patients diagnosed with head and neck types of cancer, Heukelom et al. analyzed the [^89^Zr]-cetuximab uptake in the view of chemoradiotherapy selection using either the antiepidermal growth factor receptor (EGFR) mAb cetuximab or cisplatin [159,160]. In addition, Loon et al. performed a phase I trial for patients with head and neck and lung cancer, where the [^89^Zr]-cetuximab compound (a mAb that blocks the EGFR) was used [161].

Lamberts et al. used [^89^Zr]-MMOT0530A for patients with pancreatic and ovarian cancer and concluded that the use of this compound in conjugation with an antibody drug conjugate, DMOT4039A, may present promising results for individualized antibody-based treatment guidance [133]. A year later, Bensch et al. studied [^89^Zr]-lumretuzumab’s uptake before and during the treatment (with the HER3-antibody lumretuzumab) in patients with solid tumors [146]. Additionally, the same authors evaluated the feasibility of the [^89^Zr]-atezolizumab PET imaging tracer in solid malignancies [134].

In 2020, Pandit-Taskar et al. successfully implemented first in-human imaging using ^89^Zr-IAB22M2C as a radiolabeled body against CD8+T cells in patients diagnosed with solid malignancies [151]. Previously, several studies were focused on the biodistribution and tumor accumulation/uptake of ^89^Zr-labeled compounds (e.g., [^89^Zr]-labeled cergutuzumab amunaleukin (CEA-IL2v) [162] and [^89^Zr]-GSK2849330 [145]) in patients with advanced solid tumors [160].

##### [^64^Cu]-Labeled Compounds

With a medium half-life of 12.8 h, [^64^Cu]-radionuclide has a unique decay profile [114,163], and it is used for PET diagnostic imaging [164] but also for targeted (radio)therapy. It can be easily conjugated against proteins, antibodies, and/or peptides [112,165].

In the last decade, [^60,62,64^Cu]-diacetyl-bis (N4-methylthiosemicarbazone) (Cu-ATSM)-labeled PET radiopharmaceuticals have been developed for targeting the hypoxic regions in tumors [166,167,168]. For [^64^Cu]-ATSM, one of the recently described uptake mechanisms is that Cu(II)-ATSM is reduced by thiols and, consequently, converted into the Cu(I)-ATSM complex in both normal and hypoxic conditions [169].

For head and neck squamous cell carcinomas, McCall et al. investigated the role of [^64^Cu]-ATSM (HNSCC) using a combination of in vivo PET imaging and in vitro autoradiography [166]. The results of the study indicated a significantly higher uptake of [^64^Cu]-ATSM in tumors than the uptake in muscles, therefore concluding that [^64^Cu]-ATSM uptake is specific for malignant expression [165,166]. A few studies [167,168] have assessed the prognostic significance of [^64^Cu]-ATSM in patients with locally advanced non-small-cell lung cancer or head and neck cancer prior to treatment. However, no significant differences were found between early (1 h postinjection) and late (16 h postinjection) acquisitions [165], with [^64^Cu]-ATSM showing high sensitivity but low specificity in therapy response prediction, a result that can be correlated with undetectable rates of hypoxia [167,168].

With respect to [^60^Cu]-radionuclide, the studies focused on cervical cancer and predicting the tumor response post-therapy; observed that the [^60^Cu]- and [^64^Cu]-ATSM magnitude of uptake are similar, even if [^64^Cu]-ATSM presented a better image quality; and concluded that indeed [^64^Cu]-ATSM may be a more suitable predictive indicator of tumor response to therapy in patients diagnosed with cervical cancer [165].

In rat models, there is a good correlation between low oxygen partial pressure and high [^64^Cu]-ATSM uptake. However, preclinical data suggested that [^64^Cu]-ATSM should not be considered as a hypoxia marker in all types of tumor. For instance, in a study conducted by Vāvere et al., the interplay factors between [^64^Cu]-ATSM and the overexpression of fatty acid synthase were directly correlated with prostate cancer, despite low [^64^Cu]-ATSM uptake levels caused by elevated hypoxic conditions [165].

As for other standard [^64^Cu]-based radiopharmaceuticals, [^64^Cu]-labeled monoclonal antibodies are widely used for the detection of small colorectal tumor foci in the abdomen and pelvis, and [^64^Cu]-DOTATATE was developed for neuroendocrine tumor imaging and is currently considered in several ongoing clinical trials due to its favorable pharmacokinetics and high stability [163].

In terms of novel approaches, the [^64^Cu]-PSMA compound (included in Appendix A) seems feasible for prostate cancer detection [163], [^64^Cu]-dichloride is used for copper trafficking in metabolic diseases [170,171], and [^64^Cu]-labeled peptides based on uPAR biomarkers (already developed and administrated to humans) can be used for glioblastoma [163]. Additionally, as previously stated at the beginning of this section, due to the radionuclide’s unique decay profile, ^64^Cu can also be used for internal radiotherapy since its favorable β-decay (38%) and Auger electrons emitted from this nuclide are able to damage the tumor cells [172,173,174].

##### [^68^Ga]-Labeled Compounds

Gallium has 31 known isotopes and 11 metastable isomers (the two naturally occurring stable isotopes gallium-69 (60.11%) and gallium-71 (39.89%)). The two gallium isotopes applied for nuclear medicine imaging are gallium-67, with a longest half-life of 3.26 days, and gallium-68, with a half-life of just 67.7 min [112,114,175,176]. Gallium-68 is the most widely used radionuclide, labeled against PSMA-11 and in combination with [^177^Lu]-Lu-DOTA-PSMA-617, in the diagnosis of prostate cancer with additional theranostic application, due to ^68^Ga/^177^Lu-radiolabelled tracers’ similar biological behavior [177]. Moreover, a ^68^Ga theranostic pair is also considered for neuroendocrine tumors. In this context, the most widely used analogues of somatostatin with gallium-68 are [^68^Ga]-Ga-DOTA-TOC, [^68^Ga]-Ga-DOTA-TATE, [^68^Ga]-Ga-DOTA-LAN, and [^68^Ga]-Ga-DOTA-NOC [178]. Additionally, yttrium-90 and lutetium-177 are used as therapeutic counterparts [175]. Several ^68^Ga-labeled radiopharmaceuticals based on exendin-4 (a glucagon-like protein-1 receptor agonist) have been developed, and it was shown that this type of tracers (e.g., [^68^Ga]-Ga-DOTA-exendin-4) is able to accurately localize small pancreatic tumors known to produce an excessive amount of insulin. It was also demonstrated that [^68^Ga]-Ga-DOTA-exendin-4 identifies these insulomas significantly better than ^111^in-labelled radiopharmaceuticals [179].

For the gastrin-releasing peptide receptor’s (GRPR) imaging, which is overexpressed in several types of cancer, such as prostate cancer, breast cancer, colorectal and pancreatic cancer, and lung and ovarian cancer, several gallium-68-based radiopharmaceuticals were designed [175], specifically [^68^Ga]-Ga-BBN-RGD for breast cancer [180], [^68^Ga]-NOTA-DUPA-RM26 for prostate cancer, and [^68^Ga]-Ga-NOTA-Aca-BBN for glioma imaging [181]. Similarly to metallic [^89^Zr]-radiolabeled compounds, ^68^Ga-based radiopharmaceuticals (Appendix A) are used for targeting human epidermal growth factor receptors (HER2) [182] and carcinoembryonic antigen (CEA), which is highly increased in certain types of cancer, but also in benign conditions.

#### 2.1.2. PET Radiopharmaceuticals in Neurology

Without engaging any pharmacological effects, a large variety of PET radiotracers are now considered depending on the targeted biological (in this section, neurological) conditions. Among all these conditions, the worth-noting ones are based on differentiating frontotemporal dementia in Alzheimer disease (AD) [183], developing key biomarkers that can accurately detect the receptors of interest in Huntington’s disease (HD) [184,185], differentiating other neurodegenerative disorders [186] through molecular phenotyping, diseases’ severity evaluations, seizure onset areas in epilepsy, encephalitis location and diagnosis [183], brain tumor prognosis, and tumoral extent delineations [187].

Within the central nervous system (CNS), the molecular sensitivity of the PET imaging technique enables the quantification of target-ligand interactions with great selectivity in humans, and became particularly useful in further understanding the pathologies and identifying potential new targets for therapeutic purposes. Several well-known and widely debated radiopharmaceuticals for brain imaging include [^18^F]-FDG compounds used for imaging the alterations of glucose metabolism [188,189], [^18^F]-FDOPA tracers for dopamine synthesis in schizophrenia and PD, and other tracers for translocator proteins’ detection in AD and/or PD. In addition, [^11^C]-PIB pharmaceuticals are extensively used for tracking the amyloid β plaques’ accumulation in AD [190].

In general, the passive diffusion of PET tracers into the brain must follow a lower molecular weight (<500 kDa), a lipophilic coefficient between 1 and 5 (at physiological pH), and a topological polar surface area below 90 Å^2^ [191,192,193]. In terms of pharmacokinetic properties, the essential criteria of all CNS tracers rely on their ability to cross the BBB, while the tracer’s selectivity ultimately influences its usefulness and applicability. However, other relevant parameters regarding the targets’ expression, alteration in their affinity states (high or low), internalization, and changes in endogenous occupancy must be also considered [194,195].

Since 2012, a consistent number of PET tracers have been developed and evaluated for the detection of abnormal and insoluble accumulation of misfolded proteins within brain cells, considered to be the responsible factors in AD, PD, and HD pathologies [183]. These accumulations in the human brain are known as amyloid β plaques, tau, α-synuclein, and/or fibrils [196].

Amyloid PET tracers (Appendix A) are used for the detection of pathological amyloid depositions within the brain tissue [197] (including the white matter), and are mainly benzothiazole and benzoxazole derivatives [196,198,199,200,201,202,203], such as [^18^F]-flutemetamol, [^18^F]-florbetapir, and [^18^F]-florbetaben [204,205]. In patients presenting mild cognitive impairment and AD, high cortical fibrillar amyloid loads were observed when [^18^F]-flutemetamol, [^18^F]-florbetapir, [^18^F]-florbetaben, and [^18^F]-flutafuranol compounds were used [206,207,208]. Recently, other tracers with improved binding specificity and diminished bone uptake (e.g., [^18^F]-FIBT, [^18^F]-FACT, and [^18^F]-D15FSP), and others that are able to detect diffuse amyloids (e.g., [^18^F]-fluselenamyl) were also developed [201,209,210,211]. In amyloid transgenic mouse models [200,202], several antibody-based PET/SPECT radiotracers, particularly [^124^I]-RmAb158-scFv8D3 and [^124^I]-8D3-F(ab’)2-h158, have been proposed while showing an adequate BBB entrance.

For treating affective disorders, such as depression, drugs must target the serotonergic system consisting of serotonin (5HT) G-protein-coupled-receptor families and a ligand-gated channel family [212]. Therefore, examples of PET radiotracers with distinctive affinities towards the 5HT system include [^18^F]-MPPF, [carbonyl-^11^C]-WAY-100635 for 5HT_1A_, [^18^F]-altanserin, and [^11^C]-MDL for 5HT_2A_, [^11^C]-GSK215083 for 5HT_6_ [213,214], and [^11^C]-DASB for the 5HT transporter [212].

The aggregation rates of tau proteins (negative regulators of mRNA translation) represent a pathologic hallmark of AD and are directly correlated with the disease’s progression [215]. Between 2013 and 2018, 12 PET tracers were developed, including [^18^F]-AV-1451 and [^18^F]-THK [216], being the most extensively studied [217]. Among others, the [^18^F]-AV-1451 tracer exhibits high selectivity and fast kinetics, allowing for an accurate description of the disease’s stages [218,219,220,221,222,223,224,225,226,227,228,229,230,231,232,233,234,235,236]. However, it also presents low affinity patterns for non-AD tau accumulations [230,233,234,235], restricting the tracer’s use for other tau-related neuropathies’ imaging.

As previously stated, [^18^F]-THK (523) represents another highly debated tau radiotracer, with its first implementation in humans in 2014 [236]. In both healthy controls and AD conditions, [^18^F]-THK (523)’s uptake regions are mainly described by subcortical white matter areas, with medium selectivity over amyloids. Until 2016, three other [^18^F]-THK-based PET tracers were developed and tested in human [237,238,239], showing highest uptake in cerebellar grey matter regions [237,238,240,241,242], improved properties over [^18^F]-THK (523), and better differentiation between AD and healthy controls [241,243]. Appendix A highlights additional radiotracers used for tau protein aggregate imaging, the available in vitro and ex vivo binding studies [244] indicating potential off-target bindings of in-human tau tracers [230,231,232,245,246,247], and limited data for in vivo selectivity reports [248].

Taking into consideration that many neurological disorders are directly correlated with particular dysfunctions of the neuroreceptors, transporters, and/or synaptic proteins [249,250], the imaging of the abnormal accumulation of tau filaments is also associated with different affinities of tau tracers for their distinctive binding sites [251,252]. In addition, several in vivo studies have identified a link between the cholinergic system and AD, PD (with dementia), and Lewy body pathologies; therefore, multiple potential targets for cholinergic imaging were also considered [253,254,255,256,257,258,259,260]. Examples of these tracers include [^18^F]-FEOBV, [^18^F]-VAT, and [^18^F]-ASEM with their heterogeneous distribution in line with the target distribution and fast kinetics within most brain areas [259,260,261,262,263,264,265,266,267].

Adenosine 2A receptors (A_2A_) are G-protein-coupled receptors targeted by CNS neurotransmitters and highly implicated in multiple neurological disorders [268], such as schizophrenia, bipolar disorder, HD, PD, preclinical models of AD, addiction, aging, epilepsy, and multiple sclerosis [269,270]. In this framework, the most reliable tracers developed for targeting A_2A_ are [^11^C]-TMSX and [^11^C]-SCH442416. However, the pharmacokinetic properties of these tracers, despite the fact that they can easily cross the BBB, imply nonspecific bindings in humans, with a low dynamic range of receptor occupancy [270]. To overcome this limitation, in 2015, Barret et al. [271,272] developed the [^18^F]-MNI-444 tracer with a promising characterization of the adenosine receptors across multiple regions of the brain (striatal and cerebellum areas). The [^18^F]-MNI-444 radiotracer appears superior for A_2A_ imaging in the CNS to the other evaluated tracers, presenting a higher selectivity and better pharmacokinetic properties for imaging [271,272]. It also presents higher retention in the striatum and fast washout in the cerebellum, where the concentration of A_2A_ is very low [270].

For targeting synaptic vesicle glycoprotein 2A (SV2A) [273], which is specifically expressed in synaptic terminals and particularly studied for antiepileptic treatment development, the [^11^C]-UCB-J and [^18^F]-UCB-H radiotracers have recently been proposed [274]. Both [^18^F]-UCB-H and [^11^C]-UCB-J compounds show a high degree of SV2A-specific bindings and fast kinetics [275], with [^11^C]-UCB-J presenting higher stability rates, when compared with the other one in healthy controls’ test–retest scans [276,277,278].

Other tracers targeting different types of receptors (Appendix A) include: [^11^C]-BU99008, being the only tracer for imidazoline receptors in humans with great response in blocking studies [279,280,281,282] but showing slow kinetics and therefore requiring long scan times; [^11^C]-ITMM, which shows promising performance in preclinical studies for targeting metabotropic glutamate receptor 1 [283,284,285,286,287,288], however requiring long scan times and presenting low brain uptake and slow metabolism [289]; [^11^C]-GR103545 and [^11^C]-LY2795050 tracers, have recently been tested in humans for targeting K opioid receptors [290,291,292]; and [^11^C]-Cimbi-36 tracers, target the serotonin 5-HT_2_ receptor subtype [293,294,295] with high binding selectivity and relatively high uptake levels in cortical areas, although showing moderately slow pharmacokinetic profiles [296]. Additionally, for CNS enzyme targets in humans, several new PET radiotracers were developed between 2013 and 2018, including [^11^C]-PS13 [297,298], [^18^F]-BCPP-EF [299,300,301,302,303,304,305], [^11^C]-martinostat [306,307,308], [^18^F]-MNI-654 [309], and [^11^C]-MK-3168 [310,311,312]. Further details regarding this type of tracers and many others are highlighted in Appendix A.

Lu et al. announced the discovery of [^18^F]-LSN3316612 as a novel PET ligand for measuring *O*-linked-*N*-acetyl-glucosamine (O-GlcNAc) hydrolase (OGA) in the human brain. It increases tau protein phosphorylation, which is a biomarker for AD. Moreover, the research was expanded to include PET quantification of O-GlcNAcase in the brains of healthy human volunteers using [^18^F]-LSN3316612, with V_T_ calculated using a two-tissue compartment model, as well as region-based and voxel-based quantification of [^18^F]-LSN3316612 in the human brain for O-GlcNAcase [313,314]. Lindberg et al. created the radioligand [^18^F]-AZ10419096 for imaging 5-HT1B receptors. The anomalies of the 5-HT1B receptor are linked to CNS illnesses, including anxiety and depression, and have become a great indicator of examination using PET. AZ10419096 demonstrated selectively high brain uptake into the 5-HT1B receptor in a baseline PET analysis with [^18^F]; however, in a blocking PET investigation with an AR-A000002 blocker, the binding was found to be 80% in the occipital cortex, suggesting strong specific binding [315].

The [^18^F]-MK-6240 compound, a new radiotracer for imaging neurofibrillary tangles, was described by Koole et al. Tangles are a significant pathogenic hallmark of Alzheimer’s disease that is linked to cognitive deterioration. Preclinical toxicity studies, first-in-human biodistribution, and dosimetry studies for clinical application in imaging investigations of the human brain have all validated the safety of [^18^F]-MK-6240 [316].

The new radioligand [^11^C]-MC1 was described by Shrestha et al. for imaging the low-density cyclooxygenase 2 (COX-2) enzyme in the human brain. Neuroinflammation requires it as a target. In healthy brain tissue, COX-1 expression is high, whereas COX-2 expression is low, although it is elevated during the inflammatory process. This [^11^C]-MC1 is a selective radioligand for COX-2, and it was discovered that it detected high-density COX-2 in symptomatic joints in rheumatoid arthritis patients. The particular binding to COX-2 was validated by blocking tests with celecoxib in this work, which showed significant absorption of [^11^C]-MC1 in the human brain [317].

In 2021, Yan et al. used the PET radioligand [^11^C]-deschloroclozapine ([^11^C]-DCZ) to measure the number of muscarinic DREADDs transfected in monkey brains. Due to a substantially lower background uptake, the signal-to-background ratio of [^11^C]-DCZ was about twofold larger than that of [^11^C]-clozapine ([^11^C]-CLZ) in the monkey DREADD model. As a result, [^11^C]-deschloroclozapine appears to have a higher selectivity for DREADD hM4Di receptors than [^11^C]-CLZ [318]. DREADDs (designer receptors exclusively activated by designer drugs) are a new chemogenetic technique that may be used to activate or inhibit distinct neuronal populations in the brain, allowing for the regulation of various neurological diseases [319].

The first-in-human assessment of a new radioligand, [^11^C]-PS13, to measure cyclooxygenase-1 (COX-1) in the brain was published by Kim et al. [320]. The vesicular monoamine transporter 2 (VMAT2) protein is used as a biomarker to assess PD symptoms. VMAT2 receptors transport and store the produced dopamine in the synaptic storage vesicles in the striatal cells. As a result, a recent study used micro-PET to produce a new radiotracer, 10−(+)−[^11^C]-DTBZ, for in vivo PET imaging of VMAT2 to better understand PD [321].

Even though [^18^F]-FDG is widely used and exhibits great performances in brain imaging, a research group reported a potential PET tracer with even more potential for brain glioma imaging. Thus, *N*-(2-[^18^F]-fluoropropionyl)-l-glutamate ([^18^F]-FPGLU) showed remarkable visibility of the tumor and the capability to detect tumor activity [322].

With so many tracers on development that show great feasibility, the future prospects of PET imaging in a wide range of diseases is rather encouraging. The possibility of exceeding the performances of current radiopharmaceuticals in clinical use is also to be considered; thus, further testing of the available potential agents is mandatory.

#### 2.1.3. PET Radiopharmaceuticals for Cardiovascular Events

The first application of radiotracers for myocardial blood flow studies in humans was reported in 1927. Among all nuclear cardiology techniques, myocardial perfusion imaging is nowadays the most heavily used. Nuclear cardiology has extended its implementations over the last decades to heart function, blood circulation, and noninvasive imaging of myocardial viability, cardiac inflammation, cardiac metabolism, and prognosis [323,324].

The most important properties of an ideal PET/SPECT radiotracer for myocardial perfusion imaging include a physical half-life of the radionuclides that allows sufficient time for image acquisition and repeated measurements, easy and stable labeling, optimal radiation dosimetry for integral test–retest evaluations, high myocardial uptake, linear dependence between the myocardial uptake and coronary blood flow, rapid blood clearance, and, lastly, high target-to-background ratio involving lower radiotracer uptake levels in the adjacent organs, such as lungs, stomach, liver, and bowels [323,325,326].

For the myocardial uptake assessment [327] or the myocardial blood flow measurement [328,329], [^13^N]-ammonia represents a suitable radiotracer [330,331,332], however, imposing some limitations due to its physical half-life of 10 min. Conversely, rubidium-82 has the advantage of its short half-life of only 75.5 s [333,334,335,336,337,338], allowing for complete rest–stress studies within 30 min. Moreover, it exhibits high diagnostic accuracy in the clinical detection of coronary artery disease [339]. Another promising radiotracer, an analogue of mitochondrial (complex I) inhibitors, is represented by the [^18^F]-flurpiridaz compound [340,341].

Keeping in mind that the heart acquires its energy to pump from several sources, including glucose, ketones, and free fatty acids, various radiotracers targeting these sources (see Appendix A) were investigated to study different aspects of cardiac metabolism [342]. In 2018, Manabe et al. classified in their review article the cardiac imaging radiotracers into: inorganic tracers, radiometal ions, small organic tracers, and radiometal complexes [342]. The inorganic radiopharmaceuticals, labeled with short half-life radionuclides, such as ^13^N and ^15^O [343] (10 and 2 min, respectively), have been widely used for cardiac perfusion imaging. Additionally, the radiometal positron emitter ^82^Rb^+^, presenting an ionic radius comparable to that of K^+^ in its monovalent cationic form and belonging to the same family as K (such as alkaline metals), exhibits kinetics similar to those of K^+^ and has been extensively engaged as a PET imaging tracer [344]. Other than cationic radionuclides, the monovalent anion ^18^F^−^ has been widely used for calcification lesion imaging in bone PET scans [345]. With regard to small organic tracers, ^11^C-epinephrine and ^18^F-fluorodopamine are also used to image the presynaptic sympathetic nervous system of the heart [346], since it plays an essential role in the abnormal processes associated with various cardiac diseases. In addition, in terms of cardiac inflammation [347], but also for various CNS pathologies and psychiatric diseases [348,349,350] as previously discussed in Section 2.1.2, the translocator proteins encoded by translocator protein (TSPO) genes and mainly expressed on the outer mitochondrial membrane are considered suitable targets for the radiolabeled receptor ligands in PET imaging [351]. Among the tracers used for targeting TSPO receptors, [^11^C]-PK11195 is used for carotid stenosis imaging and shows a great potential for detecting inflammatory macrophages in atherosclerotic plaques as well [352]. Additionally, several next-generation TSPO-targeted radioligands have been developed to improve the binding specificity and to lower the signal-to-noise ratio for neuroinflammation imaging, subsequently enhancing the cardiovascular inflammation imaging performance [349,350]. Here, other tested radiolabeled compounds include [^18^F]-FEDAA1106 [347], [^18^F]-FEMPA [353], and [^18^F]-GE180 [354].

The approved PET tracers for myocardial perfusion imaging and their molecular targets include [^82^Rb]-chloride targeting Na^+^/K^+^ ATPase cotransporters (the most commonly used), [^13^NH_3_]-ammonia [355] with a retention in the myocardium mediated by the ATP-dependent conversion of glutamate to glutamine, and the traditional H_2_ ^15^O [356,357], which is not metabolically retained in tissues but used as a standard validation against which other tracers are being evaluated for optimal myocardial blood flow quantification [358].

For targeting mitochondrial complex I and the mitochondrial membrane, research regarding novel PET tracers has been oriented towards ^18^F radioisotope due to its relatively long half-life of 110 min, enabling tracer synthesis, but short positron range, allowing for higher image spatial resolutions [358]. Within this framework, the novel ^18^F-labeled tracers are classified considering two mechanisms of mitochondrial targeting: (1) targeting mitochondrial complex I through analogues, MC1 inhibitors, which specifically bind to the MC1 receptors on the inner mitochondrial membrane, and (2) targeting mitochondrial membrane considering lipophilic cations that are able to penetrate it through electrical potential voltage gradients [359,360]. Within the first targeting mechanism group, the PET tracers include [^18^F]-flurpiridaz [340,341,361,362], [^18^F]-FPTP2 [363,364], and [^18^F]-FDHR [358], whereas from the second targeting mechanism group, novel PET tracers include [^18^F]-FBnTP, [^18^F]-FTPP [358], and [^18^F]-FPTP [365].

In 2020, Heo et al. published a review article dedicated to current and novel PET/SPECT radiopharmaceuticals specifically used to image cardiovascular inflammatory diseases [366]. They stated that imaging cardiovascular inflammation presents several challenges, from the small sizes of atherosclerotic lesions and complex pathophysiological characteristics of atherosclerotic plaques, to targeting the specificity and sensitivity of the required radiotracers and their pharmacokinetic properties. However, PET imaging techniques are preferred owing to their high sensitivity, quantitative and functional detections, noninvasive nature, and well-established approaches for human translation [366,367,368]. Going back to the great importance of glucose metabolism that is significantly enhanced in activated inflammatory cells, glucose transporters such as GLUT-1 and GLUT-3 [369], among many blood cell types, such as monocytes, macrophages (M0, M1, and M2), neutrophils, B cells, T cells, and platelets, are known to lead to higher accumulation of metabolized glucose analogues, including the gold standard [^18^F]-FDG [358,370].

The chemokine receptors are also considered promising targets for imaging inflammatory diseases, such as atherosclerosis and its corresponding complications. The alpha-chemokine CXCR4 receptor [371] acts as a biomarker of atherosclerosis in both humans and animal models. Several studies have demonstrated that CXCR4’s specific uptake of [^68^Ga]-pentixafor is associated with calcified plaques of carotid stenosis and other cardiovascular risk factors, proving the radiotracer’s potential for further evaluation of atherosclerotic lesions. The same studies also demonstrated the CXCR4’s upregulation within rabbit and human plaques, mainly colocalized with macrophage staining through immunohistochemistry [372,373,374,375,376]. Moreover, [^68^Ga]-pentixafor PET images of postmyocardial infarction patients presented significant differences in the radiotracer’s uptake, designating different degrees of inflammatory responses. The tracer’s uptake might also be able to predict heart failure in postmyocardial infarction patients and to detect culprit plaques within the coronary arteries.

The [^64^Cu]-DOTA-ECL1i radiotracer is another compound widely used for cardiovascular inflammation and is also being investigated as an imaging agent in clinical trials for imaging lung inflammation [377,378] and pancreatic cancer. In addition, in mouse models of atherosclerotic plaque, [^64^Cu]-DOTA-ECL1i shows decreased levels of monocytes and macrophages, playing as an indicator of inflammatory leukocyte migration to plaque regression [379], therefore being able to image leukocytes in vivo [380]. Other ^64^Cu-radiolabeled compounds, including [^64^Cu]-DOTA-DAPTA and [^64^Cu]-DOTA-DAPTA-comb, which has a conjugated polymeric structure, were tested in a femoral artery injury of atherosclerosis-prone apolipoprotein E-deficient (apoE−/−) mice model, showing high uptake levels in injured lesions. However, the second tracer, a nanoparticle-based one, showed even higher accumulation rates when compared with the first tracer [381]. Moreover, in the same mice model, in vivo blocking studies using [^64^Cu]-DOTA-vMIP-II demonstrated that more than eight chemokine receptors, including CCR1, CCR2, CCR3, CCR4, CCR5, CCR8, CXCR4, and CX3CR1, contributed to the uptake [382].

With respect to [^68^Ga]-labeled radiopharmaceuticals (Appendix A), several preclinical studies showed specific bindings of [^68^Ga]-DOTATATE and [^68^Ga]-DOTANOC compounds to macrophages originated from blood monocytes in the atherosclerotic plaques of mouse aorta [383,384]. Studies focused on [^68^Ga]-DOTATATE and [^64^Cu]-DOTATATE tracers also showed increased uptakes in coronary arteries and large arteries [385,386], reporting a direct correlation between these radiotracers’ uptake and cardiovascular risk factors’ assessment, but with a potential higher uptake performance for [^64^Cu]-DOTATATE [387,388].

Another radiotracer using DOTATATE peptide as vehicle molecule, the [^177^Lu]-DOTATATE compound was approved for the radionuclide therapy of gastroenteropancreatic neuroendocrine tumors and showed that its use during the treatment significantly reduced the [^68^Ga]-DOTATATE accumulation in plaques, encouraging this particular therapy against inflammatory conditions related to plaques [389].

The [^18^F]-2-fluoro-2-deoxy-d-mannose ([^18^F]-FDM) tracer presents similar in vivo and ex vivo uptakes to ^18^F-FDG in atherosclerotic rabbits, however, imposing much higher in vitro uptake levels into macrophages, contrasting ^18^F-FDG [390]. A probable explanation for this might involve [^18^F]-FDM’s uptake mechanism, since mannose—the sugar monomer of the aldohexose series of carbohydrates—is taken up by macrophages through the protein family GLUTs; therefore, [^18^F]-FDM total uptake depends not only on GLUTs but also on macrophage mannose receptors, being able to provide additional insights when compared with ^18^F-FDG. However, it might also mean that they could share the same PET imaging limitations of [^18^F]-FDG.

Another tracer with higher standardized uptake values in abdominal aortas of atherosclerotic rabbits, when compared with ^18^F-FDG, is the [^68^Ga]-labeled NOTA-neomannosylated human serum albumin ([^68^Ga]-NOTA-MSA) [391]. Moreover, several [^68^Ga]-labeled macrophage mannose receptor antibodies or nanobodies were studied in atherosclerotic mouse and/or rabbit models, showing higher accumulation rates in aortas, when compared with healthy control models. In 2018, Senders et al. concluded that the uptake levels of [^68^Ga]-labeled macrophage mannose receptors, together with [^18^F]-FDG and [^18^F]-NaF, exhibit different inflammatory cell populations and uptake mechanisms [392]. Moreover, a year before, Varasteh et al. targeted mannose receptor expression in macrophages on atherosclerotic plaques of apolipoprotein E-knockout mice using tilmanocept compound labeled with [^111^In], and observed that [^111^In]-tilmanocept accumulated in mannose receptor-expressing organs, such as liver and spleen, but associated with only low residual blood signal. The same study concluded that a tilmanocept-radiolabeled compound might represent a promising tracer for noninvasive detections of macrophages in atherosclerotic plaques [393].

Hellberg et al. conducted in 2016 a retrospective imaging study using the PET tracer [^18^F]-fluoromethylcholine ([^18^F]-FMCH) and demonstrated that type 2 diabetes enhances the arterial uptake of choline in atherosclerotic mice models, when compared with nondiabetic models, even though both models showed comparable macrophage content [394]. The study also provides diagnostic advantages for choline transportation imaging in patients diagnosed with type 2 diabetes, since PET imaging with [^18^F]-FDG can be quite challenging due to elevated blood glucose levels in those patients. In the same year, Vöö et al. demonstrated the feasibility and potential of the [^18^F]-FCH tracer to evaluate the vulnerable plaques and to differentiate them from the stable ones [395]. The [^18^F]-FLT is another PET radiopharmaceutical compound that showed higher uptake levels not only in atherosclerotic plaques of mice and rabbit models but also in humans. However, when compared with [^18^F]-FDG, [^18^F]-FLT showed no significant myocardial uptakes [396], but uptakes in other proliferating cells (parenchymal cells) might affect [^18^F]-FLT’s specificity, considering that endothelial and smooth muscle cells also contain plaques. 

Last but not least, other PET tracers widely used or recently developed for cardiovascular imaging or to monitor treatments’ efficacy include [^89^Zr]-DNP [397], [^18^F]-macroflor [398], [^111^In]-DANBIRT [399,400], and [^18^F]-FB-IL2 [401]. Since matrix metalloproteinases (MMPs)—or matrixins—represent a family of proteinases with multiple and crucial roles in the inflammatory processes, imaging the MMPs could provide insights not on the density of macrophages, like most probes, but on their activity to promote inflammation and plaque ruptures. In this context, the [^111^In]-DTPA-RP782-radiolabeled MMP inhibitor has been considered for molecular imaging of atherosclerosis and aneurysm and, particularly, monitoring of distinctive MMP activities [402,403,404,405]. More preclinical and clinical studies are mandatory to further assess its feasibility; nevertheless, the development of these selective MMP inhibitors is already in progress [406,407,408] in order to minimize the side effects of treatments.

#### 2.1.4. PET Radiopharmaceuticals for Bacteria Imaging

Taking into account the heterogeneity of animal models and bacterial strains, Auletta et al. classified the research studies focused on bacteria PET imaging (Appendix A) based on the type of bacterium (Gram-positive, Gram-negative, Gram-positive and negative, others) [409]. According to the literature, most studies that used Gram-positive bacteria imaging obtained better results in terms of radiopharmaceutical binding in both animal models and humans. Ubiquicidin peptide fragments, specifically UBI-31-38 and UBI-29-41, are reported to have enabled the imaging of *Staphylococcus aureus* (*S. aureus*) infection foci in models with high target-to-nontarget ratios. However, even though the preliminary results showed safety in terms of toxicity rates in humans, further research must be conducted [410,411,412,413].

The ^124^I-labeled FIAU compound reported good results in animal models; however when translated to humans, the results were less promising [414]. [^18^F]-FDG-6-P agents are potential substrates for bacterial hexose phosphate transporters, heavily expressed in many bacteria species, and present the ability to discriminate between infection and sterile inflammation; thus further research might confirm their initial potential as a new radiopharmaceutical for bacteria imaging [415]. Compared with [^18^F]-FDG, [^18^F]-FDG-6-phosphate shows similar biodistributions both visually and semiquantitatively. However, in contrast with [^18^F]-FDG, [^18^F]-FDG-6-phosphate indicates no accumulation in sterile inflammation of noninfected mice models [415].

The radiolabelled ciprofloxacin showed high specificity when labeled with gallium-68 [416], but interestingly, when labeled with technetium-99m [417] or fluorine-18 [409], the results led to nonspecific patterns. Their divergent results involved no apparent specific interactions, and the uptake using both radionuclides was rapidly cleared. Note that it is worth concluding that the difference between the used isotopes and the labeling chemistry might affect the biological behavior of radiopharmaceuticals [409]. Most studies conducted using Gram-positive bacteria reported promising results when radiolabeled glucose analogues were used, particularly for [^18^F]-FDS compounds that showed high binding specificity to *E. coli* or *K. pneumoniae* [418,419,420].

Other sensitive and specific radiopharmaceuticals for *E. coli* detection include [^18^F]-fluoromaltose and [^18^F]-FAG [421,422]. When using both Gram-positive and Gram-negative bacteria or a combination of bacteria to induce infection in the preclinical models, [^18^F]-fluoropropyl-trimethoprim, [^18^F]-fluoromaltose, and other fluorine-labeled analogues showed good specificity for bacterial infection imaging with possible applicability in clinics [423,424,425]. Although [^68^Ga]-gallium-DOTA-TBIA101 is also considered a promising agent for *E. coli* infection [426], when the infection is being induced by *S. aureus* or *M. tuberculosis*, this radiopharmaceutical was found to be nonspecific [427].

#### 2.1.5. PET Radiopharmaceuticals for Infection/Inflammation

Inflammation is associated with a variety of human diseases, including stroke, Alzheimer’s disease, atherosclerosis, autoimmune diseases, and even malignant conditions, either directly or indirectly. As a result, information gleaned from molecular imaging of inflammation in various conditions is extremely useful for disease diagnosis and prognosis, therapeutic response monitoring, and better understanding disease processes. Many inflammation-related biomarkers, such as inflammatory cell metabolism, membrane markers, cytokines, and vascular alterations during inflammation, have been discovered and explored for potential imaging or therapeutic targets thus far. However, not many such radiopharmaceuticals have been developed in recent years.

In both physiological and pathological settings, B cells are essential for controlling immunological responses. The deregulation of B-cell function is assumed to be the cause of a number of B-cell-mediated illnesses, such as B-cell malignancies (e.g., lymphomas, leukemia) [428], autoimmune disorders (e.g., rheumatoid arthritis (RA)) [429], multiple sclerosis (MS) [430,431,432], inflammatory illnesses (e.g., obesity, diabetes [433,434]), and transplant complications [435]. Thus, B cells are a very important therapeutic target. BTK is a cytoplasmic tyrosine kinase that is expressed by both B cells and myeloid cells such as microglia. The kinase is implicated in various signal transduction pathways that are important in B-cell development [436]. BTK inhibitors are being studied for the treatment of B-cell malignancies [437], but also in order to combat inflammatory and autoimmune illnesses [438], such as RA [439] and MS [440]. Radiolabeling BTK inhibitors and using the resulting radiopharmaceuticals in PET imaging will be extremely beneficial in the monitoring and treatment of B-cell-mediated diseases.

Donnelly et al. proposed the syntheses of such radiopharmaceuticals, using carbon-11 isotopes, due to their optimal half-life of 20 min. By the end of the study, one of the radiopharmaceuticals presented encouraging prospects as a novel PET tracer. [^11^C]-ibrutinib presented >98% radiochemical purity and molar activity of 7.6 ± 2.7 Ci/μmol (281 ± 99 Gbq/μmol) and with a half-life ranging from 19.89 to 20.15 min. Moreover, quality control tests revealed that this radiopharmaceutical is sterile [441].

In peripheral tissues, TSPO is widely expressed, but in the healthy human brain, it is only moderately expressed. TSPO expression was shown to be high in macrophages, neutrophils, lymphocytes [352,442], activated microglia, and astrocytes in previous research [443,444]. Most investigated radiotracers used in PET imaging are either ^11^C or ^18^F labeled. Examples of such tracers that bind to TSPO are: ^11^C or ^18^F-labeled isoquinoline carboxamide PK11195 (1-(2-chlorophenyl)-*N*-methyl-*N*-(1-methylpropyl)-3-isoquinoline carboxamide) [352] and, more recently, ^11^C-PBR28 (*N*-(2-[^11^C]-methoxybenzyl)-*N*-(4-phenoxypyridin-3-yl)acetamide) [443,445]. Moreover, it has been found that in rat models, TSPO ligands were able to accumulate in infarct areas. Furthermore, PET imaging using TSPO has also been reported to have shown promising results in atherosclerosis detection, ^3^H-DAA1106 ((*N*-5-fluoro-2-phenoxyphenyl)-*N*-(2,5-dimethoxybenzyl)acetamide), ^3^H-(R)-PK11195, and ^11^C-PK11195 being such examples [352]. PET imaging using the TSPO radio-ligands ^18^F-FEDAC, ^11^C-(R)-PK11195 41, and ^123^I-(R)-PK11195 57 revealed considerable lung lesion absorption, mostly from active neutrophils and macrophages, in a lipopolysaccharide-induced infectious lung inflammation model [442].

The somatostatin receptor (SSTR) has been taken into account for neuroendocrine tumor imaging. However, very few PET tracers have been reported to target SSTR, but it has been deemed more successful in SPECT imaging [446].

MMPs are zinc- and calcium-dependent metalloproteases that may destroy extracellular matrix (ECM) protein components. MMPs and their inhibitors, MMPIs, regulate the balance of extracellular proteolysis, and increased MMP activity is linked to cancer, atherosclerosis, and other inflammatory diseases. In a separate investigation [402], ApoE-/- mice with a high-cholesterol diet were given ^18^F-MMPI. MMP-positive plaques may be seen in the aorta’s inner curvature using ex vivo PET/CT.

Interleukin-2 (IL-2) is a 133-amino-acid single-chain glycoprotein produced and released by activated T cells, particularly CD4^+^ and CD8^+^ Th1 lymphocytes. Many forms of inflammatory disorders, such as inflammatory degenerative diseases, transplant rejection, tumor inflammation, organ-specific autoimmune diseases, and adipose inflammatory insulin resistance, are associated with T-cell activation. Di Gialleonardo et al. described the production of ^18^F-FB-IL-2 by tagging IL-2 with *N*-succinimidyl 4-^18^F-fluorobenzoate (^18^F-SFB) to identify activated T cells in inflammation [401]. These preliminary findings imply that ^18^F-FB-IL-2 is stable, physiologically active, and capable of detecting activated T cells in vivo [447].

VCAM-1, a member of the immunoglobulin superfamily of endothelial adhesion molecules, is a vascular cell adhesion molecule. It is crucial in all phases of atherosclerotic plaque formation [448]. It is found on active endothelium and can cause macrophage adherence in the early stages of plaque development. VHPKQHR, a linear peptide affinity ligand, was discovered in apolipoprotein E-deficient mice utilizing in vivo phage display. This peptide is similar to very late antigen-4, a known VCAM-1 ligand. Based on this peptide sequence, a multivalent PET imaging agent (^18^F-4V) was created and used to assess VCAM-1 expression. PET imaging of atherosclerotic plaques in the aortic root in (ApoE)-/- mice revealed a large focused signal in the aortic root [448].

The [^18^F]-FDG was found to be the most suitable radiopharmaceutical for infectious and inflammatory diseases. For fungal infections, this radiotracer showed promising results by detecting pathophysiological changes in earlier stages and being able to image infectious foci. Moreover, [^18^F]-FDG is a good candidate for biopsy, as it is able to detect most active infection sites. Studies also confirmed that this radiopharmaceutical is suitable for treatment monitoring and regulation [449,450,451]. It has been also reported that [^18^F]-FDG has a great impact on large vessel vasculitis (LVV) imaging and treatment recommendations. Specifically, a study reported a 26.7% change in the treatment of patients with LVV who did not receive immunosuppressive medication and a 22.6% change of treatment in those who received medication [452]. In order to prevent younger patients’ exposure to radiation, [^18^F]-FDG-PET/MRI showed great feasibility in LVV detection in preliminary results; however, further research must be conducted in order to confirm its efficacy [453].

Current applications of FDG-PET include bloodstream infections of unknown origins, fever of unknown origins, infective endocarditis, vascular graft infections, spondylodiscitis, and cyst infections [454]. Another research article suggested that [^18^F]-FDG is better for monitoring the disease during immunosuppressive therapy. However, when it comes to imaging an immunological network in cancer, [^18^F]-FDG is not suitable, particularly in the early phases of the treatment. The radiopharmaceutical targets the tumor cells as well as inflammatory cells; thus [^18^F]-FDG might show an increase in metabolic burden due to the fact that immunotherapeutic drugs present an inflammatory response. Other studies also confirmed the [^18^F]-FDG tracer’s inability to differentiate from pseudo-progression and when a patient should cut off therapy [455,456].

The [^64^Cu]-GTSM is widely used for imaging inflammation using cell migration trafficking [457] and expected to be more suitable than [^64^Cu]-PTSM. Moreover, in a review article published in 2019, Werry et al. presented an overview of recent developments in TSPO-dependent PET imaging biomarkers for neuroinflammation in neurodegenerative disorders [458]. The inflammatory responses in the brain and spinal cord are known to promote the activation of microglia and astrocytes, and represent a pathological hallmark in many central nervous system diseases. A direct correlation between activated microglia, inflammatory cytokines, and TSPO ligand bindings has been demonstrated in neurodegenerative patterns of animal models; therefore, increased TSPO PET signals may occur in disease-relevant brain regions within a large variety of neurodegenerative disorders [458,459]. For monitoring this type of inflammations in neurodegenerative diseases, [^11^C]-PK11195 [460,461], [^11^C]-ER176 [460,462], [^18^F]-GE-180 [463,464,465,466,467,468,469,470], [^11^C]-DPA-713 [460,465], [^18^F]-DPA-713 [471], [^11^C]-PBR-28 [460,462,472,473,474,475,476,477,478,479], [^18^F]-PBR111 [480,481], [^18^F]-FEDAA1106 [482,483], and [^11^C]-DAA1106 [484] radiotracers have been investigated.

#### 2.1.6. Novel PET Tracers in Oncology and PET Radiopharmaceuticals with Proven Use in Clinical Practice

In oncology, several studies reported novel radiotracers for PET imaging. For example, an ^18^F-labeled tropomyosin receptor kinase (TRK) inhibitor, [^18^F]-TRACK, was described by Gauthier et al. Overexpression of the TRK A, B, or C receptors leads to tumor development, metastasis, and neurological disorders, such as Alzheimer’s and Parkinson’s. In experiments with nonhuman primates, [^18^F]-(R)-TRACK was observed to pass the BBB and demonstrated high brain uptake [485]. Mossine et al. described an ^18^F-labeled sulfonamide analogue as a human carbonic anhydrase IX inhibitor for PET imaging of hypoxic tumors [486].

Song et al. used ^18^F-IRS as an EGF-specific ligand to image mutant EGF receptors in patients with NSCLC. Overproduction of EGF receptor ligands in the tumor microenvironment causes severe EGFR receptor alterations, which speed up epithelial tumor cell proliferation, invasion, and metastasis [487,488]. In vivo, this radiotracer exhibited excellent tumor cell absorption by targeting mutant EGF receptors.

Several studies have demonstrated the overexpression of TSPO in gliomas, and a recent research, conducted by Su et al., found that using the [^11^C]-PK11195 radiotracer, low-grade astrocytomas and oligodendrogliomas may be clearly distinguished using a PET dynamic analysis [489]. PET molecular imaging studies with endogenous reporter genes have been translated into clinical research, and one recent study with trimethoprim labeled with carbon-11 ([^11^C]-TMP) in a xenograft mouse model showed high biodistribution and good sensitivity toward *Escherichia coli* dihydrofolate reductase expressing cells, implying that it could provide advancement in current PET reporter gene technologies [490].

Since the absorption of [^18^F]-fluciclovine in prostate cancer cells was shown to be high in patients with recurrent prostate cancer [491], [^18^F]-fluciclovine, also known as Axumin, has been authorized by the US Food and Drug Administration (FDA) for clinical use. Another novel ^18^F-labeled tracer was reported by Strebl et al. Their work revealed significant brain uptake, specific binding, and regional distribution of the novel radiotracer [^18^F]-MGS3 for PET imaging of histone deacetylase in humans [492]. Moreover, among other tracers with clinical approval, a transgenic mouse model of breast cancer study revealed encouraging findings using [^11^C]-PAQ to monitor the cancer treatment status and was deemed a a good radiotracer in cancer therapy response treatment [493]. Last but not least, the [^11^C]-4DST (4′-[methyl-11C]-thiothymidine) compound, a thymidine derivative, has also been authorized for clinical use, particularly for the assessment of cancerous cell development, metastasis, and invasion [494,495].

### 2.2. SPECT Radiopharmaceuticals

The ^99m^Tc is a pure gamma emitter radionuclide with a half-life of 6 h. It decays into ^99^Tc by isomeric transition (140.5 keV 98.6% and 142.6 keV 1.4%). It is readily available from ^99^Mo/^99m^Tc generators and is extracted as a [^99m^Tc(VII)]-O_4_ solution. The physical properties for ^99m^Tc are close to ideal for SPECT and planar scintigraphy imaging: the 140.5 keV photon energy is optimal for gamma cameras, being a pure gamma emitter ensures no additional radiation burden to the patient, and the versatility of ^99m^Tc chemistry allows for the development of a multitude of easy-to-prepare compounds through the introduction of cold labeling kits. Technetium-labeled radiopharmaceuticals are the most used imaging agents in nuclear medicine worldwide and cover a wide range of applications: thyroid, bone, renal, myocardial perfusion, intestinal, cerebral, hepatic, cerebral imaging, and so forth [9,496,497].

In the last decade, a large number of studies were devoted to technetium chemistry and developing new radiopharmaceuticals; however, the imaging capabilities of SPECT systems in terms of sensitivity and resolution were outperformed by PET/CT. While the 1990s and 2000s saw a significant number of new radiopharmaceuticals transitioning to clinical practice, in the last decade, just a few passed the clinical trials. Nevertheless, the last years saw a renewed interest in SPECT imaging through the introduction of CZT detector technology, more specialized dedicated systems, and advances in image reconstruction and quantification [498,499,500,501,502,503].

Modern technetium-labeled radiopharmaceuticals are developed through indirect labeling using bifunctional chelators. An important aspect of this method is finding the ideal chelator to incorporate the radiometal and link it to the pharmacophore. An interesting feature of ^99m^Tc is the formation of characteristic moieties that dictate the chemical properties of the resulting complexes. Examples of technetium cores used in the development of promising compounds are [^99m^Tc(I)]-tricarbonyl, [^99m^Tc(V)]-HYNIC, and [^99m^Tc(V)]-nitrido [10]. To this end, a large body of work was directed to the synthesis and structural and chemical characterization of bifunctional chelating agents for technetium.

#### 2.2.1. SPECT Radiopharmaceuticals in Oncology

SPECT imaging in clinical oncology plays an important role in identifying various tumors overexpressing particular receptors. Radiopharmaceuticals have been successfully developed for imaging of somatostatin receptors (neuroendocrine, gastroenteropancreatic, breast, brain, and small cell lung cancer tumors), prostate-specific membrane antigen (prostate cancer), gastrin-releasing peptide receptor (prostate, breast, pancreas, small cell lung cancer, and colorectal tumors), melanocortin 1 receptor (melanomas), and integrin α_ν_β_3_ receptor (brain, lung, ovary, breast, and skin cancer), to name a few [504].

SSTR-targeting Tc-labeled radiopharmaceuticals were the first to be developed. The first commercially available agent was [^99m^Tc(V)]-EDDA-HYNIC-TOC, consisting of a [^99m^Tc(V)]-HYNIC core, EDDA as coligand, and Tyr3-octreotide as an SSTR agonist peptide [10,505]. Studies showed good specificity and sensitivity and performed similarly in a direct comparison with [^111^In]-DTPA-octreotide. The [^99m^Tc(V)]-HYNIC/EDDA-TOC is a reasonable alternative, the main advantages being related to the superior physical properties of [^99m^Tc] as an imaging agent [506].

The choice of agonist peptides as pharmacophores was natural, considering the increase in tumor uptake due to internalization; however, it was determined that RFs with peptide antagonists had higher uptake most likely due to the fact that antagonists bind to multiple sites, increasing radiopharmaceutical retention [10]. Makris et al. developed a series of compounds using the tricarbonyl core and NOTA and NODAGA as chelators, conjugated to a SSTR type 2 antagonist peptide. Structural characterization was performed on the Re analogues by means of LC-ESIMS, HR-ESI-MS, and NMR spectroscopy. Radiolabeling resulted in high yields (>95%) determined by HPLC, and both were stable in rat serum with a protein binding of 9%–24% [507]. In the preclinical studies, biodistribution and SPECT imaging were evaluated in AR42J tumor-bearing ICR SCID mice. [^99m^Tc(I)]-NOTA revealed significantly higher tumor uptake than [^99m^Tc]-NODAGA, 16.7 ± 3.32%ID/g compared with 2.78 ± 0.27%ID/g. Both complexes had medium to low tumor retention, rapid renal clearance, and tumor accumulation, which could clearly be identified on SPECT/CT images in accordance with biodistribution data [508].

Another recent study [509] compared two ^99m^Tc radiolabeled somatostatin receptor antagonists based on a new peptide, SS-01 (p-Cl-Phe-cyclo(D-Cys-Tyr-D-Trp-Lys-Thr-Cys)D-Tyr-Nh2), using the HYNIC and tetraamine chelators. Preclinical evaluation revealed a better performance from the tetraamine compound [^99m^Tc(V)]-N4-SS-01 with a very high tumor uptake of 47%/IA/g and rapid clearance from nonspecific organs, resulting in high-contrast SPECT/CT images, while the HYNIC compound exhibited almost no uptake in vitro. As a follow-up, Fani et al. [510] evaluated two [^99m^Tc(V)]-tetraamine sst2 antagonists: TECANT-1 based on an LM3 structure and TECANT-2 based on the SS-01 peptide. The preclinical results in HEK-SST2-bearing mice revealed similar uptake and stability. TECANT-1 had lower uptake in blood, kidney, and muscles, providing the best tumor to background ratios, and was selected for clinical translation. Although the [^99m^Tc(V)]-tetraamine compounds performed better, Gaonkar et al. [511] investigated why the Tc-HYNIC fragment lost tumor cell uptake when conjugated with the SS-01 antagonist peptide. They evaluated the effects of spacers of different lengths on [^99m^Tc(V)]-HYNIC/EDDA conjugated with the potent antagonist peptides SS-01 and JR11 in a head-to-head comparison with [^99m^Tc(V)]-N4-SS-01 and [^99m^Tc(V)]-N4-JR11, with the commercially available [^99m^Tc(V)]-HYNIC/EDDA-TOC as reference. In vitro studies determined that introducing the spacer increased cellular uptake depending on the spacer’s length. The best uptake was found for the Ahx (C_6_ amino-hexanoic acid) spacer with a 16-fold increase compared with [^99m^Tc(V)]-HYNIC/EDDA-SS-01, nearly on the same level as the reference. Further biodistribution and SPECT/CT studies of [^99m^Tc(V)]-HYNIC/EDDA-Ahx-SS-01 and [^99m^Tc(V)]-HYNIC/EDDA-Ahx-JR11 in HEK-SST2-tumor-bearing mice revealed a similar picture. Although tumor uptake was still higher for the N4 conjugates, they also exhibited increased uptake in the kidneys and higher background activity. This, in turn, resulted in higher contrast SPECT/CT images for the HYNIC conjugates.

PSMA has become the primary focus of targeted imaging and therapy in prostate cancer. A great number of PSMA inhibitor small-molecule imaging agents have been developed for both SPECT/CT and PET/CT [10,512,513]. The most representative are [^123^I]-MIP-1072, [^99m^Tc(I)]-MIP-1404, [^99m^Tc(V)]-HYNIC-iPSMA, and [^99m^Tc(V)]-O-N_3_S-PSMA I&S.

The [^123^I]-MIP-1072 and ^123^I-MIP-1095 based on the Glu-urea-Lys motif were the first to gain clinical attention. Preclinical results showed higher tumor uptake for MIP-1072 at 1 h postinjection and higher uptake for MIP-1095 at 24 h postinjection. Based on these results MIP-1072 was further evaluated as an imaging agent for SPECT/CT clinical studies, while MIP-1095 was proposed as a therapeutic agent labeled with ^125/131^I due to higher tumor retention [514,515,516].

Hillier et al. [517] evaluated four technetium-labeled PSMA inhibitors based on the [^99m^Tc(I)] (CO)_3_^+^ core and Glu-urea-Glu and Glu-urea-Lys pharmacophores conjugated to CIM or TIM chelators. Out of the four, [^99m^Tc(I)]-MIP-1404 showed the most promising results. In phase I and II clinical trials, the compound successfully identified small lesions in the prostate gland, lymph nodes, and bones [517,518]. However, it failed the specificity coprimary endpoint of a phase III clinical trial, detecting clinically meaningful prostate cancer with a specificity of 71–75% [513].

Promising results using the [^99m^Tc(V)]-HYNIC core were obtained for [^99m^Tc(V)]-HYNIC/EDDA-iPSMA and [^99m^Tc(V)]-HYNIC/EDDA-ALUG [519,520,521,522,523]. The [^99m^Tc(V)]-HYNIC/EDDA-ALUG showed good binding affinity to PSMA-positive LNCaP cells and small animal imaging of mice bearing LNCaP tumor xenographs [519]. The [^99m^Tc(V)]-HYNIC/EDDA-iPSMA composed of a ligand incorporating the Lys(NaI)-urea-Glu inhibitor was synthesized through kit formulation with high yield and no need for further purification. Results showed high tumor uptake, mainly kidney excretion, with preliminary SPECT/CT images able to detect tumors and metastasis in patients with prostate cancer [520,521]. Follow-up studies [522,523] evaluated the performance of [^99m^Tc(V)]-HYNIC/EDDA-iPSMA SPECT/CT in comparison with [^68^Ga]-PSMA-11 PET/CT images based on maximal standardized uptake values on prostate, bone lesions, and lymph nodes. The authors found the performance of [^99m^Tc(V)-HYNIC/EDDA-iPSMA comparable to that of [^68^Ga]-PSMA-11 for prostate and bone lesions and for lymph nodes with a size greater that 10 mm; however, the technetium compound had lower sensitivity for small lesions, detecting only 28% of the nodes with a size of less than 10 mm, identified by the PET scan. This can be attributed to the lower sensibility and resolution of SPECT systems and the technical difficulty in SPECT SUV calibrations.

The [^99m^Tc(V)]-O-N_3_S-PSMA-I&S [524,525] was designed for radio-guided surgery. The [Tc(V)≡O]^3+^ core was coordinated to a mercaptoacetyl-trisine chelator coupled to the PSMA targeting pharmacophore. Biodistribution studies in patients showed slow clearance, which allowed imaging at 3-5 h after injection for visualization of PSMA-rich lesions. Various hospitals implemented protocols for radio-guided surgery and reported successful intraoperative detection and removal of metastatic lesions.

A different method for PSMA receptor targeting is based on utilizing radiolabeled nanobodies. Nanobodies are fragments of antibodies that retain certain binding properties and exhibit faster pharmacokinetics than the original structure [526].

Evazalipour et al. [527] developed a series of technetium-labeled anti-PSMA nanobodies. The two most promising compounds, [^99m^Tc(I)]-PSMA30 and [^99m^Tc(I)]-PSMA6, showed specific binding affinity for LNCaP but not PC3 cells. PSMA30 had better tumor uptake and tumor-to-normal-organ ratio and was chosen for further investigations.

Several novel α_ν_β_3_-targeting SPECT imaging agents have been developed based on the RGD motif (arginyl-glycyl-aspartic acid). However, so far, none have been introduced in clinical practice. The [^99m^Tc]-maraciclatide ([^99m^Tc]-NC100692) is a cyclic peptide conjugated to the RGD motif initially evaluated in breast and lung cancer. Recently, Cook et al. [528] investigated the potential of [^99m^Tc]-maraciclatide in the detection and therapy response assessment in patients with prostate cancer. They successfully demonstrated that 99mTc-maraciclatide specifically accumulates in metastatic bone lesions in patients with prostate cancer. Furthermore, based on measurements of CT Hounsfield units, they determined a possible relation between the uptake and the degree of osteoclast activity.

The [^99m^Tc(V)]-3P4-RGD2 [529,530,531,532], a promising compound based on a HYNIC chelator, PEG4 linkers, and RGD2 peptide, showed good pharmacokinetic properties. Clinical investigations showed good results for the detection of malignant lesions in pulmonary cancer [529] and bone metastasis [533]. Two other studies comparing [^99m^Tc(V)]-3P4-RGD2 with [^99m^Tc(I)]-sestamibi scintimammography revealed that [^99m^Tc(V)]-3P4-RGD2 had slightly better performance in sensitivity, specificity, and accuracy, but it was not statistically significant [530,531]. Two other prospective studies performed comparison studies of [^99m^Tc(V)]-3PRGD2 with [^18^F]-FDG in breast cancer [534] and esophageal cancer [535] showing comparable results between the two compounds, highlighting the potential of this technetium radiopharmaceutical.

#### 2.2.2. SPECT Radiopharmaceuticals for Cardiovascular Events

Similarly to PET radiotracers, myocardial perfusion imaging with SPECT plays a pivotal role in cardiology, predominantly in the diagnosis and risk assessment of coronary artery disease (CAD). There are three SPECT imaging agents currently in clinical use: [^201^Tl]-Cl, [^99m^Tc(I)]-sestamibi, and [^99m^Tc(V)]-tetrofosmin. The [^201^Tl]-Cl is used less than the Tc analogues due to limitations regarding long half-life, attenuation artifacts produced by the low characteristic X-ray emission in the range of 69 to 81 keV, and administered dose constraints due to radiation burden [536].

The [^99m^Tc(I)]-sestamibi is a lipophilic cation consisting of a technetium core bound to six 2-methoxyisobutylisonitrile (MIBI) ligands. Given the importance of this radiopharmaceutical, we previously reported a detailed structural characterization of the MIBI ligand by means of vibrational spectroscopy and density functional theory calculations [537]. The [^99m^Tc(V)]–tetrofosmin consists of a di-oxo core bound to two diphosphine ligands. Both compounds benefit from easy cold kit preparation and were developed in the 1980s. The two tracers perform similarly with some differences regarding extraction coefficient, heart uptake, blood and liver washout, and optimal time of imaging from injection [538].

In a comprehensive review conducted by Boschi et al. [539], the authors highlight the fact that despite the extensive use of these compounds in nuclear cardiology for decades, they do not meet the requirements of an ideal perfusion imaging agent. The main disadvantages are low first-pass extraction, high liver absorption which can hinder the interpretation of the lower and left ventricular wall, and lack of a linear relationship between uptake and coronary blood flow.

Promising imaging agents were developed based on the [^99m^Tc][Tc(V)N(L)(PNP)]^+^ scaffold consisting of the [^99m^Tc]-nitride core [^99m^Tc(V)≡N]^2+^, a PNP-type bis-phosphine, and one monoanionic chelate (L) [540]. Biodistribution studies of [^99m^Tc(V)]-N-MPO, consisting of a [^99m^Tc(V)≡N]^2+^ core with 2-mercaptopyridine and biphosphine as chelators, showed higher myocardial uptake than [^99m^Tc(I)]-sestamibi and very fast liver clearance. Clinical studies showed favorable results. Due to the fast liver washout, the myocardium was clearly separated from the left liver lobe on images acquired at 10 min after injection, making [^99m^Tc(V)]-N-MPO a good candidate for imaging patients with known CAD [541].

In a more recent study, Salvarese et al. [542] evaluated a new class of complexes of the general formula [^99m^Tc][Tc(V)N(DASD)(PNP*n*)]^+^, where DASD = 1,4-dioxa-8-azspiro[4,5]-decandithiocarbamate and PNP*n* = bisphosphinoamine. The compounds are modified versions of the [^99m^Tc(V)]-N-DBODC(5) complex, which showed comparable results to commercially available agents in clinical studies. Animal studies showed increased myocardial uptake and superior heart-to-liver ratios for two of the four compounds investigated when compared with [^99m^Tc(V)]-N-DBODC(5), [^99m^Tc(I)]-sestamibi, and [^99m^Tc(V)]-tetrofosmin.

The [^99m^Tc(III)]-teboroxime is one of the first approved tracers for cardiac imaging. It has excellent pharmacokinetic properties: high first pass extraction, linear relation between uptake and blood flow and rapid clearance [543,544,545]. The main drawback of this tracer is the very fast cardiac washout that leaves a 5 min window for image acquisition, which is difficult to achieve on conventional gamma cameras. A new class of [^99m^Tc(III)] derivatives was developed [546,547,548] similar in structure to teboroxime except for the replacement of the methyl group added to the boronate cap by a sulfonyl group. As a follow-up, Xi et al. [549] performed a comparative study on normal and acute myocardial infarction swine between [^99m^Tc(III)]-3SPboroxime and [^99m^Tc(I)]-sestamibi. The results showed high initial heart uptake, longer myocardial retention, and high heart-to-background ratio. These results make [^99m^Tc(III)]-3SPboroxime an excellent candidate for future clinical studies.

#### 2.2.3. SPECT Radiopharmaceuticals in Neurological Disorders

The most commonly used SPECT imaging agents in neurology are [^99m^Tc(V)]-HMPAO, [^123^I]-ioflupane. and [^99m^Tc(I)]-TRODAT-1. The [^99m^Tc(V)]-HMPAO ([^99m^Tc(V)]-exametazime) is regularly used for the assessment of cerebral perfusion in neurological diseases [550,551]. [^123^I]-ioflupane and [^99m^Tc(I)]-TRODAT-1 bind to the presynaptic dopamine transporter to give images of the striatum. They are useful in the diagnosis of Parkinson’s disease, which can be evaluated by the asymmetric or degraded uptake in the two striata. Although the preparation of [^99m^Tc(I)]-TRODAT-1 is more convenient, its brain uptake is lower than that of [^123^I]-ioflupane [550].

Promising ^123^I-based tracers have been developed for beta-amyloid imaging. Chen et al. developed a series of imidazopyridine compounds [552]. The most successful compound based on brain permeability, rapid nonspecific organ washout, and affinity to Aβ aggregate, termed DRM106 (6-iodo-2-[4-(1H-3-pyrazolyl)phenyl]imidazole[1,2-α]pyridine), was further evaluated in vivo. Results showed that the performance of the trace is comparable to that of [^11^C]-PiB in detecting Aβ deposition. In 2016, Maya et al. [553] evaluated a novel tracer, [^123^I]-I-ABC577, in preclinical and clinical studies. The results revealed favorable pharmacokinetics, and SPECT imaging was able to differentiate Alzheimer’s disease patients from normal controls.

One of the limiting factors of designing technetium radiopharmaceuticals for neurological applications is their poor ability to cross the blood–brain barrier. Many different compounds have been designed that bind to Aβ plaques, but they do not mark all the checks required to pass the barrier [554].

Sangou et al. [555] reported novel technetium/rhenium derivatives of benzothiazole, benzimidazole, and *N*-methylbenzimidazole that demonstrated affinity for Aβ plaques and impressive brain uptake. The structures of the compounds were obtained by replacing the phenyl moiety of 2-phenylbenzothiazole with the cyclopentadienyl tricarbonyl core [Cp^99m^Tc(I)(CO)_3_]^+^. The 2-arylbenzothiazole scaffold has been extensively used as a pharmacophore for Aβ plaques, and the [Cp^99m^Tc(I)(CO)_3_]^+^ possesses favorable properties for BBB penetration. The Re complexes were characterized by IR, NMR, MS-ESI, and elemental analysis. X-ray crystallography was possible for the Re benzothiazole derivative. Spectroscopic analysis revealed a piano-stool-type structure. The [^99m^Tc] derivatives obtained were stable with high radiolabeling yield and retained binding affinity for Aβ plaques and showed impressive brain uptake of 7.94 ± 1.46, 3.99 ± 0.6, and 5.36 ± 0.65%IA/g at 2 min postinjection. Overall, these tracers show remarkable properties for Aβ imaging [555].

## 3. Materials and Methods

A literature research was conducted within scientific databases, including PubMed, Scopus, and Web of Science, and random google searches. Our interest focuses on articles and/or review papers published in the last 10 years. Two authors screened and selected (independently) the studies related to the herein presented topics on PET, and two authors were (independently) focused on the literature related to radiopharmaceuticals used for SPECT scans.

All searches were based on multiple combinations of the following keywords: “PET/CT” OR “positron emission tomography/computed tomography”, “radionuclide”, “radiopharmaceuticals”, “ligands” OR ”radioligand”, “PET tracers”, “oncology”, “overview”, “SPECT/CT” OR “single photon emission computed tomography/computed tomography, “Technetium radiopharmaceuticals”, “SPECT radiopharmaceuticals”, “myocardial perfusion imaging”, and “structural characterization”.

For the studies reported in this review article, the authors considered gathering the required information within the following parameters: author, year of publication, articles’ DOI (for an easier access to the articles of interest), title, radionuclide, half-life, compound, disease, uptake localization, function, standard/new, key aspects, short abstract, and conclusions.

Moreover, in order to complete the herein final list of references, the authors also considered the reference lists of the primary and relevant studies.

## 4. Conclusions

### 4.1. PET Radiopharmaceuticals

Based on our findings within the available literature, there are encouraging results that suggest great development in the radiopharmaceutical industry for PET imaging. Among the most effective PET tracers with a wide range of applications are the ^18^F- and ^11^C-labeled radiopharmaceuticals due to their convenient half-life. Although there are few tracers developed in the past 10 years regarding inflammatory diseases, many novel tracers have been released to clinical use and subjected to oncological and neurological pathologies. Moreover, many tracers have shown great preliminary results, and even though further research is needed for their validation and commercialization, the prospects are promising.

### 4.2. SPECT Radiopharmaceuticals

Significant effort has made in researching novel SPECT radiopharmaceuticals, with limiting results, especially in comparison with PET tracers. The few that were evaluated in clinical trials were found to be statistically similar to currently used first-generation radiopharmaceuticals. Due to advances in the chemistry of bifunctional chelators and the versatility of [^99m^Tc] metal fragments, promising results have been published in more recent literature. The bulk of the research has focused on peptide receptor targeting radiopharmaceuticals for tumor imaging, specifically for SSTR and PSMA targeting. Compounds with favorable pharmacokinetics were obtained using a variety of ligands, most notably NOTA/NODAGA, HYNIC, and tetramine, which were successfully conjugated to relevant peptides, all showing good tumor uptake and tumor-to-background ratios in preclinical evaluations.

Moreover, advances in SPECT detector design demonstrating higher resolution, sensitivity, and faster acquisition times have renewed interest in older radiopharmaceuticals, such as [^99m^Tc(III)]-teboroxime, prompting the development of new derivatives with excellent potential for myocardial perfusion imaging. Although the preclinical results are promising, further clinical studies are needed for the validation of biodistribution data and imaging capabilities in human patients for these novel compounds. Nevertheless, we conclude that SPECT radiopharmaceuticals still play a significant role in molecular imaging as a complementary technique to PET imaging.

## Figures and Tables

**Figure 1 ijms-23-05023-f001:**
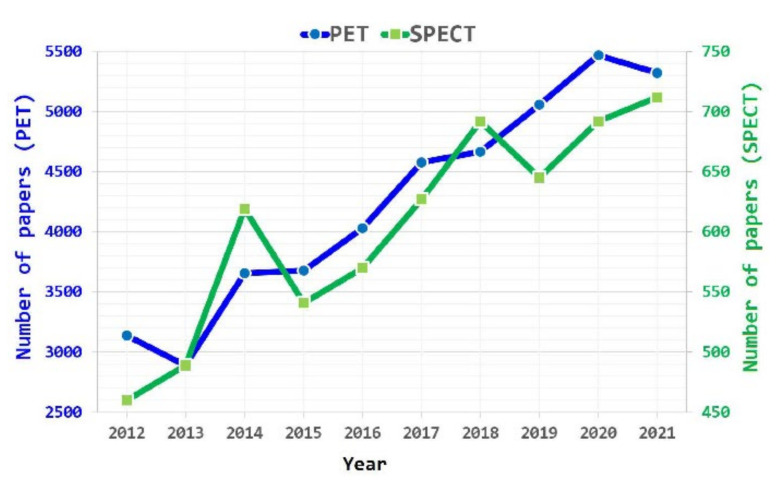
Number of scientific papers related to PET (left axis) and SPECT (right axis) radiopharmaceuticals over the past 10 years.

## Data Availability

No new data were created or analyzed in this study. Data sharing is not applicable to this article.

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
