# Peer review of "Radiopharmaceuticals for PET and SPECT Imaging: A Literature Review over the Last Decade"

_ijms, 2022, doi:10.3390/ijms23095023_

Round 1

Reviewer 1 Report

The authors have put a lot of hard work into this paper! It is an important overview of radiotracers available for both PET & SPECT, a detailed description of their chemical properties, advantages and limitations as well as their clinical indications. 

General Comments:

  1. The medical terminology has to be improved and correspond to routinely used wording and phrases.
  2. Add a section regarding to radiation exposure in general & for specific tracers and doses.
  3. Reorganize the subsections starting with introductory comments followed by a clear   structure that should repeat itself for each of the radioisotopes and for each of the tracers. 
  4. Give more emphasis to tracers with proven use in clinical practice.

Author Response

First of all we thank the editor and the reviewers for considering our manuscript.

Pertinent comments and suggestions made by reviewers, as well as their recommendations regarding the manuscript’s organization, general aspects of radiation exposure, medical terminology, chemical structures, and a proper list of references were extremely useful for improving our work. We considered every single comment and suggestion raised by reviewers and made the required changes. The changes in the manuscript are highlighted in yellow. Please note that we have included a graphical abstract of the manuscript, and we have inserted two new references (ref. [51] and [542]).

In the following, the reviewers’ comments are written in italic type, our point-by-point answers in normal font type and the bold type was used for emphasizing the changes within the revised manuscript.

Reviewer #1

Comments and Suggestions for Authors

The authors have put a lot of hard work into this paper! It is an important overview of radiotracers available for both PET & SPECT, a detailed description of their chemical properties, advantages and limitations as well as their clinical indications.

General Comments:

  1. The medical terminology has to be improved and correspond to routinely used wording and phrases.

Answer: Thank you for pointing out the errors in our medical terminology. We have re-evaluated the manuscript and paid more attention to an up-to-date medical terminology, spelling, and abbreviations.

Therefore, the following changes include:

- page 3, line 98: “viable myocardium” was replaced with myocardial viability;

- page 5, line 223: “glioblastoma multiform” was replaced with glioblastoma;

- page 5, line 223: “hepatocarcinomas” was replaced with hepatocellular carcinoma (HCC);

- page 5, line 268: we made the required changes on HCC abbreviations;

- page 7, line 347: “granulomatoses disease” was replaced with chronic granulomatous disease (CGD);

- page 8, line 456: definition of anti-PSCA abbreviation was included (prostate stem cell antigen);

- page 9, lines 494 and 500: just the PSMA abbreviation (being already declared on page 8, line 444).

  1. Add a section regarding to radiation exposure in general & for specific tracers and doses.

Answer:

We included on page 4, line 175, the following paragraph:

As with all medical applications that use ionizing radiation, the benefit of PET and SPECT procedures must be evaluated considering the risks to patient. Dose optimization takes into consideration the administration of the amount of radioactivity that provides images of sufficient quality so as to achieve the relevant clinical information while maintaining the lowest possible radiation dose to the patient. There are different aspects that must be taken into account when deciding the administered dose such as individual patient physiology and anatomy or the design of the imaging equipment used for the procedure [11]. Average effective doses for nuclear medicine procedures range from 0.3 to 20 mSv with SPECT having generally lower effective doses than PET, mainly due to the physical characteristics of the radioisotopes used. For example, the average effective dose for a [99mTc(I)]-sestamibi cardiac rest-stress test (2-day protocol) with an administered activity of 1500 MBq is 12.8 mSv, while for a cardiac [18F]-FDG PET scan with an administered activity of 740 MBq, the average effective dose is 14.1 mSv [12]. Hybrid systems increase the radiation exposure by the addition of a CT scan. The additional radiation dose depends on whether the CT scan is used for attenuation correction, localization or for diagnostic acquisitions [13].

  1. Reorganize the subsections starting with introductory comments followed by a clear  structure that should repeat itself for each of the radioisotopes and for each of the tracers.

Answer: After careful consideration, we have decided to keep the subsections of the manuscript as they are. The main reason behind this decision is that for the tracers considered for neurological diseases, cardiovascular events, infection/inflammation and bacteria imaging, the number of used isotopes, and consequently the number of tracers, is significantly lower than the ones we found for oncological diagnosis and/or treatment. Due to the abundance of radiopharmaceutical compounds used in oncology, it was rather reasonable to present the available literature considering the isotope-related subsections.    

  1. Give more emphasis to tracers with proven use in clinical practice.

Answer: We made a few changes in the 2.1.6 section, emphasizing the novel PET tracers used in oncology and, as suggested, we highlighted the tracers widely used (and approved) in clinical practice (page 22, lines 1103-1114).

Thus, the title of the section 2.1.6 has been changed to:

2.1.6. Novel PET tracers in oncology and PET radiopharmaceuticals with proven use in clinical practice

and the following paragraph has been added at the end of the section:

Since the absorption of [18F]-fluciclovine in prostate cancer cells was shown to be high in patients with recurrent prostate cancer [496], [18F]-fluciclovine, also known as Axumin, has been authorized by the US Food and Drug Administration (FDA) for clinical use. Another novel 18F-labeled tracer has been reported by Strebl et al. Their work revealed significant brain uptake, specific binding, and regional distribution of the novel radiotracer [18F]MGS3 for PET imaging of histone deacetylase in humans [497]. Moreover, among other tracers with clinical approval, a transgenic mouse model of breast cancer study revealed encouraging findings using [11C]PAQ to monitor the cancer treatment status and deemed as a good radiotracer in cancer therapy response treatment [498]. Last but not least, [11C]4DST (4′-[methyl-11C]thiothymidine) compound, a thymidine derivative, has been also authorized for clinical use, particularly for the assessment of cancerous cells development, metastasis and invasion [499, 500].

For SPECT radiopharmaceuticals it is hard to emphasize on those with proven use in clinical practice. In the past decade most are still evaluated in preclinical trials. We believe that in the next few years we will see a rise in clinically relevant novel SPECT tracers.

Reviewer 2 Report

The article reviewed describes a recent overview of radiopharmaceuticals for SPECT and PET imaging over the last decade. It is an impressive work with over 560 references reported. In my opinion, the review meets the required objectives for such a work, a clearly written, up-to-date but more understandable document for specialists in the field than for non-specialists. Nevertheless, this review could be interesting and useful for young PhD students starting research in this field of radiopharmaceuticals.

Before being accepted for publications, different points must be precised ad/or added and/or corrected.

1-Page 1, lines 44,45,46:

"Two of the most important advantages of PET over SPECT modality are represented by the PET’s higher sensitivity and the less challenging tracers’ synthesis methods making PET a versatile and powerful tool for clinical and research applications“.

The authors should clarify the meaning of "less challeging tracers’synthesis" for PET tracers than for SPECT tracers. I disagree on this point. For example, the synthesis of 64Cu tracers is not so easy and Tc(I) tracer syntheses are not so difficult anymore!

2- In the whole review, no chemical structure (for the most classical chelating ligands) is (re)presented. I think that a table (even in the supplementary data) gathering the most classical ligands for the most important PET and SPECT radionuclides would be useful for young readers.

3- As I mentioned earlier, the list of references is impressive. As a result, there are some errors or inconsistencies. For example:

- Check the journal abbreviation for all references (i.e. references 177; 179-180 where the journal name is in full).

- Line 464, P.10 the authors cited "according to Yoon et al." for references 111-113. This is the wrong author, please correct.

- The review is about the last decade. Given the large number of references, I recommend removing all references before 2012 (i.e. ref 510, 535, 544, 547...).

4- For the technetium-99m part, the degree of oxidation of technetium-99m must be specified for each 99mTc radiopharmaceutical.

5- On "2.1.6 Molecular PET Imaging for Drug Development. General Aspects and Novel PET 1065 Tracers in Oncology", the authors have developed from line 1106 to 1131 different aspects to be considered for the synthesis of future PET tracers. These criteria are not very new and have already been used by most of the authors whose work is referenced in this review. I do not see the relevance of this section. Similarly, there is no similar comment for the synthesis of SPECT tracers.

This part should be re-written and similar comments added for the SPECT part.

Typo errors:

P.4, line 159 and P. 21, line 1074, indicate the mass number in superscript for various radionuclides,

P.24, line 1245 bone lesions

In conclusion, this manuscript merits to be published after taking into account the suggestions above-mentioned.

Author Response

First of all we thank the editor and the reviewers for considering our manuscript.

Pertinent comments and suggestions made by reviewers, as well as their recommendations regarding the manuscript’s organization, general aspects of radiation exposure, medical terminology, chemical structures, and a proper list of references were extremely useful for improving our work. We considered every single comment and suggestion raised by reviewers and made the required changes. The changes in the manuscript are highlighted in yellow. Please note that we have included a graphical abstract of the manuscript, and we have inserted two new references (ref. [51] and [542]).

In the following, the reviewers’ comments are written in italic type, our point-by-point answers in normal font type and the bold type was used for emphasizing the changes within the revised manuscript.

Reviewer #2

Comments and Suggestions for Authors

The article reviewed describes a recent overview of radiopharmaceuticals for SPECT and PET imaging over the last decade. It is an impressive work with over 560 references reported. In my opinion, the review meets the required objectives for such a work, a clearly written, up-to-date but more understandable document for specialists in the field than for non-specialists. Nevertheless, this review could be interesting and useful for young PhD students starting research in this field of radiopharmaceuticals.

Before being accepted for publications, different points must be precised ad/or added and/or corrected.

1-Page 1, lines 44,45,46: "Two of the most important advantages of PET over SPECT modality are represented by the PET’s higher sensitivity and the less challenging tracers’ synthesis methods making PET a versatile and powerful tool for clinical and research applications“.

The authors should clarify the meaning of "less challenging tracers’ synthesis" for PET tracers than for SPECT tracers. I disagree on this point. For example, the synthesis of 64Cu tracers is not so easy and Tc(I) tracer syntheses are not so difficult anymore!

Answer: We agree with the reviewer. Our intent was to emphasize the flexibility and robustness of PET tracers, given the fact that PET radioisotopes such as C, O, N, F are common in molecules of biomedical interest, it is possible to obtain a radiolabeled version of a biomolecule by direct substitution, without changing its pharmacokinetic properties which can be very useful for applications in drug development.

We propose rewording the phrase (on page 1, line 44) to: “Two of the most important advantages of PET over the SPECT modality are represented by the PET’s higher sensitivity and more robust and flexible tracers, making PET a versatile and powerful tool for clinical and research applications.

2- In the whole review, no chemical structure (for the most classical chelating ligands) is (re)presented. I think that a table (even in the supplementary data) gathering the most classical ligands for the most important PET and SPECT radionuclides would be useful for young readers.

Answer: As suggested, we have included in the supplementary material another table (Table S6) that illustrates the chemical structures of the most representative ligands and chelating agents discussed in the herein manuscript.

3- As I mentioned earlier, the list of references is impressive. As a result, there are some errors or inconsistencies. For example:

- Check the journal abbreviation for all references (i.e. references 177; 179-180 where the journal name is in full).

Answer: Thank you for the remarks! We modified the list of references accordingly. All changes are illustrated in the manuscript (in the reference section) with red.

- Line 464, P.10 the authors cited "according to Yoon et al." for references 111-113. This is the wrong author, please correct.

Answer: The Yoon et al. reference is indicated at the end of the paragraph (with  initial ref. [109]; now ref. [113]). The other references (initial [111-113], [114-121], etc., and now modified) within the same paragraph, indicate the (other) articles that discussed those particular compounds. We understand that this framework might be confusing, therefore we made a slight change as follows:

“According to Yoon et al. [113], among all synthesized 89Zr-labeled antibodies, trastuzumab [115-117] is the most frequently employed antibody, followed by bevacizumab [118-125], cetuximab [126-128], and rituximab [129, 130]...[…]...[113].”

- The review is about the last decade. Given the large number of references, I recommend removing all references before 2012 (i.e. ref 510, 535, 544, 547...).

Answer: As suggested, we removed most references before 2012, however we propose keeping references 544, 552, 553 as they are relevant to the context of specific paragraphs in the text. Furthermore by removing references 547, 548, 549, 550, 551, we replaced the old text (page 25, lines 1291-1306 with the following text:

“Promising imaging agents were developed based on the [99mTc][Tc(V)N(L)(PNP)]+ scaffold consisting of the [99m Tc]-nitride core [99m Tc(V)≡N]2+, a PNP type bis-phosphine and one monoanionic chelate (L) [545]. Biodistribution studies of [99m Tc(V)]N-MPO, consisting of a [99m Tc(V)≡N]2+ core with 2-mercaptopyridine and biphosphine as chelators, showed higher myocardial uptake than [99m Tc(I)]-sestamibi and very fast liver clearance. Clinical studies showed favorable results. Due to the fast liver washout, the myocardium was clearly separated from the left liver lobe on images acquired at 10 minutes after injection making [99m Tc(V)]N-MPO a good candidate for imaging patients with known CAD [546].

In a more recent study, Salvarese et. al. [547] evaluated a new class of complexes of general formula [99m Tc][Tc(V)N(DASD)(PNPn)]+, where DASD = 1,4-dioxa-8-azspiro[4,5]decandithiocarbamate and PNPn = bisphosphinoamine. The compounds are modified versions of the [99m Tc(V)]N-DBODC(5) complex which showed comparable results to commercially available agents in clinical studies. Animal studies showed increased myocardial uptake and superior heart to liver ratios for two of the four compounds investigated when compared to [99m Tc(V)]N-DBODC(5), [99m Tc(I)]-sestamibi and [99m Tc(V)]-tetrofosmin.

4- For the technetium-99m part, the degree of oxidation of technetium-99m must be specified for each 99mTc radiopharmaceutical.

Answer: We have updated the manuscript text and Table S5 from the supplementary material to specify the degree of oxidation of 99mTechnetium.

5- On "2.1.6 Molecular PET Imaging for Drug Development. General Aspects and Novel PET 1065 Tracers in Oncology", the authors have developed from line 1106 to 1131 different aspects to be considered for the synthesis of future PET tracers. These criteria are not very new and have already been used by most of the authors whose work is referenced in this review. I do not see the relevance of this section. Similarly, there is no similar comment for the synthesis of SPECT tracers.

This part should be re-written and similar comments added for the SPECT part.

Answer: The main purpose of the 2.1.6 section (page 21) was to emphasize the novel PET tracers used in diagnosis and treatment assessment for different types of cancers. Therefore, according to the reviewer's suggestion, we have eliminated the general aspects of PET tracers’ synthesis, and reoriented our interest towards the clinically approved compounds.

Typo errors:

P.4, line 159 and P. 21, line 1074, indicate the mass number in superscript for various radionuclides,

Answer: Thank you for the remark! We modified lines 159 and 1074 accordingly.

P.24, line 1245 bone lesions

Answer:  We made the required change on line 1245.
